# Non-canonical functions of UHRF1 maintain DNA methylation homeostasis in cancer cells

Kosuke Yamaguchi [1] ✉, Xiaoying Chen[1], Brianna Rodgers[1], Fumihito Miura [2], Pavel Bashtrykov[3], Frédéric Bonhomme [4], Catalina Salinas-Luypaert [5], Deis Haxholli[6], Nicole Gutekunst[3], Bihter Özdemir Aygenli [7], Laure Ferry[1], Olivier Kirsh [1], Marthe Laisné[1], Andrea Scelfo [5], Enes Ugur[6], Paola B. Arimondo[4], Heinrich Leonhardt [6], Masato T. Kanemaki [8,9,10], Till Bartke [7], Daniele Fachinetti [5], Albert Jeltsch [3], Takashi Ito [2] & Pierre-Antoine Defossez [1] ✉

DNA methylation is an essential epigenetic chromatin modification, and its maintenance in mammals requires the protein UHRF1. It is yet unclear if UHRF1 functions solely by stimulating DNA methylation maintenance by DNMT1, or if it has important additional functions. Using degron alleles, we show that UHRF1 depletion causes a much greater loss of DNA methylation than DNMT1 depletion. This is not caused by passive demethylation as UHRF1-depleted cells proliferate more slowly than DNMT1-depleted cells. Instead, bioinformatics, proteomics and genetics experiments establish that UHRF1, besides activating DNMT1, interacts with DNMT3A and DNMT3B and promotes their activity. In addition, we show that UHRF1 antagonizes active DNA demethylation by TET2. Therefore, UHRF1 has non-canonical roles that contribute importantly to DNA methylation homeostasis; these findings have practical implications for epigenetics in health and disease.

DNA methylation is an essential epigenetic mark in mammals. The methylation of cytosines, mostly in the CpG context, ensures the proper regulation of imprinted and tissue-specific genes, silences repeated elements, and contributes to the function of key functional elements of the genome such as centromeres[1].

The DNA methylation pattern observed in mammalian tissues is the result of a dynamic process. First, most of the cytosine methylation brought by the gametes is erased in early development, in a process that involves passive demethylation, as well as active demethylation by the TET enzymes[2]. Then, the proper tissue- and cell-specific methyl marks are re-established in the embryo starting at the time of implantation. This re-establishment of DNA methylation depends on "de novo" methyltransferases, of which there are three in muridae[3], but only two in humans: DNMT3A and DNMT3B[1].

Even after cells have acquired their proper DNA methylation pattern, the overall stability of this pattern depends on a dynamic equilibrium of gains and losses of cytosine methylation. There can be local losses of DNA methylation due to TET activity, compensated by de novo DNA methylation, in the cell types that do express DNMT3A or DNMT3B. In addition, there is a global remodeling of DNA methylation

[1]Université Paris Cité, CNRS, Epigenetics and Cell Fate, Paris, France. [2]Department of Biochemistry, Kyushu University Graduate School of Medical Sciences, Fukuoka, Japan. [3]Institute of Biochemistry and Technical Biochemistry, Department of Biochemistry, University of Stuttgart, Stuttgart, Germany. [4]Institut Pasteur, Université Paris Cité, Epigenetic Chemical Biology, CNRS, UMR 3523, Chem4Life, Paris, France. [5]Institut Curie, PSL Research University, CNRS, UMR 144, Paris, France. [6]Faculty of Biology and Center for Molecular Biosystems (BioSysM), Human Biology and BioImaging, Ludwig-Maximilians-Universität München, Munich, Germany. [7]Institute of Functional Epigenetics, Helmholtz Zentrum München, Neuherberg, Germany. [8]Department of Chromosome Science, National Institute of Genetics, Research Organization of Information and Systems (ROIS), Mishima, Shizuoka, Japan. [9]Graduate Institute for Advanced Studies, SOKENDAI, Mishima, Shizuoka, Japan. [10]Department of Biological Science, The University of Tokyo, Bunkyo-ku, Tokyo, Japan. ✉e-mail: yamako0801@icloud.com; pierre-antoine.defossez@cnrs.fr

at the time of DNA replication. Indeed, at this point, the two parental strands of DNA carrying cytosine methylation are separated, and each is used as a template for the synthesis of a daughter strand, which is initially totally devoid of cytosine methylation. It follows that every CpG that was symmetrically methylated before replication becomes hemimethylated. The process whereby the hemimethylated sites return to a fully methylated state is called "maintenance DNA methylation", and it involves two key actors: DNMT1 and UHRF1[4].

The first crucial participant in maintenance DNA methylation is the enzyme DNMT1[1]. Unlike the de novo methyltransferases, DNMT1 is expressed in every replicating cell, and it has higher DNA methyltransferase activity on hemimethylated than on unmethylated sites. This specificity of DNMT1 comes in part from intramolecular inhibitions, which have to be lifted for the enzyme to come into action[5]. Some of the molecular mechanisms contributing to lifting this inhibition after DNA replication have been uncovered, and they involve the protein UHRF1[6,7].

UHRF1 has an SRA domain that binds DNA with a preference for hemimethylated CpGs[8]. It also has a Tandem Tudor Domain (TTD) which, together with the adjoining PHD domain, binds histone H3K9me2/3[9]. In addition, the TTD domain binds an H3K9me3-like motif within DNA Ligase 1 (LIG1), which ligates Okazaki fragments on the lagging strand[10,11]. These different interactions contribute to the recruitment of UHRF1 to replicating chromatin, where it can then modify histones. Its Ubiquitin-Like (UbL) domain cooperates with its RING finger[12,13], which then targets histone H3 for mono-ubiquitination at two positions, H3K18 and H3K23[14]. The H3K18Ub/K23Ub then binds with high affinity to the RFTS domain of DNMT1, relieving the auto-inhibition[15]. In a similar fashion, UHRF1 also mono-ubiquitinates the PCNA-associated factor PAF15, which can then bind the RFTS, freeing the catalytic domain of DNMT1[16]. In addition to the H3-mediated and PAF15-mediated activation of DNMT1 by UHRF1, there is also direct physical contact between the two proteins[17,18], leading to the activation of DNMT1[19,20]. To summarize, there is incontrovertible evidence that UHRF1 is an upstream activator of DNMT1, yet these advances leave some important questions open.

One such question is whether UHRF1 controls DNA methylation only by acting on DNMT1, or whether it also impinges on other epigenetic actors. Besides its importance for the biology of normal cells, this question is especially relevant for cancer. Indeed, the DNA methylation pattern of cancer cells has characteristic abnormalities, marked by global hypomethylation and focal hypermethylation[21], and these abnormalities are likely caused, at least in part, by imperfect DNA methylation maintenance[22]. In parallel, most tumors express high levels of UHRF1[23], overexpression of UHRF1 is oncogenic[23], and UHRF1 is necessary for colon cancer cells to maintain their DNA methylation pattern and survive[24,25]. Therefore, UHRF1 is a key regulator of the cancer epigenome, and it is important to elucidate its role, both for basic research and for medical purposes.

Therefore, the questions we address in this paper are: how does UHRF1 control DNA methylation in human cancer cells? Does it only stimulate DNMT1 or does it have other functions? If yes, which one(s)?

The model we choose to investigate the question in is colorectal cancer, a prevalent disease in which the contribution of epigenetic is solidly established. Earlier studies have yielded valuable information[24,26], but some of their conclusions have suffered from technical limitations. In particular, the loss-of-function approaches have been imperfect: siRNA has effects that are asynchronous, limited in time, and sometimes partial; shRNA can be partial or select for cells with the least depletion; constitutional knock-outs can lead to adaptation; whereas inducible knock-outs have delayed kinetics. In contrast, degron alleles have emerged as very powerful tools for loss-of-function studies, permitting rapid, total, and synchronous depletion of proteins of interest in cells[27].

We generate and validate degron alleles of UHRF1 and/or DNMT1 in human colorectal cancer cell lines. We then use genomics and bioinformatics to precisely describe the DNA demethylation dynamics in these cells, leading to the conclusion that UHRF1 maintains DNA methylation in cancer cells not only by stimulating DNMT1. Proteomics and genetics lead us to conclude that UHRF1 regulates DNMT3A, DNMT3B and TET2 activity in addition to regulating DNMT1. The tools we develop will be valuable for future research efforts, and our results advance our understanding of cancer epigenetics, with potentially important therapeutic applications.

## Results

### Establishment of degron alleles for UHRF1 and DNMT1 in colorectal cancer cell lines
To investigate the respective roles of DNMT1 and UHRF1 in cancer cells, we chose as a model the human colorectal cell lines HCT116 and DLD1, as they have been widely used to study the genetic and epigenetic events that cause and sustain transformation. Both lines have an activated KRAS and microsatellite instability but maintain a near-diploid karyotype. HCT116 cells have functional p53, whereas DLD1 cells have mutated p53[28].

In these cells, we utilized the Auxin-Inducible Degron (AID) system to perform precisely controlled, rapid, and synchronous loss-of-function experiments. To prevent unwanted degradation of the target proteins in basal conditions, we employed HCT116 with a doxycycline-inducible OsTIR1[27], while we used the recently optimized F74G variant of OsTIR1 in the DLD1 background[29].

Using Cas9-mediated knock-in, we introduced the tags into the endogenous UHRF1 and/or DNMT1 genes in the HCT116 and DLD1 cell lines, and both genes simultaneously in HCT116 (Fig. 1A, B and Supplementary Fig. 1A). As UHRF1 can be inactivated by N-terminal modifications[12,13], we inserted the AID tag at the C-terminus along with the green fluorescent protein, mClover (Fig. 1A). In contrast, N-terminal tagging of DNMT1 can be used to generate a degron allele[30]. For this reason, we placed the AID tag at the N-terminus of DNMT1, accompanied by the red fluorescent protein mRuby2 (Fig. 1A). Three independent clones were generated for each construct and used in further experiments (Fig. 1C and Supplementary Fig. 1B).

### Characterization and validation of the tagged cell lines
Having obtained the lines of the desired genotypes, we then characterized them by growth assays, microscopy, and DNA methylation measurements. In the absence of auxin, the UHRF1-AID, DNMT1-AID, or UHRF1-AID/DNMT1-AID cells grew indistinguishably from the parental HCT116 or DLD1 cells (Supplementary Fig. 1C–E). We next examined the localization of tagged UHRF1 and DNMT1. In fixed cells, both proteins were nuclear with some colocalizing foci (Fig. 1D). A correlation analysis showed that UHRF1 and DNMT1 colocalize more often than randomly expected, and also that they colocalize with the heterochromatin marker H3K9me3 (Supplementary Fig. 1F). In live-cell microscopy, we found, as expected, that DNMT1 and UHRF1 had a dynamic nuclear distribution and formed colocalizing foci during S phase (Supplementary Movies 1–3).

We further verified the functionality of the tagged proteins by measuring DNA methylation levels in HCT116 derivatives by 3 independent methods: a restriction-enzyme-based assay (LUMA), liquid chromatography followed by tandem mass spectrometry (LC-MS/MS), and whole genome bisulfite sequencing (WGBS). These data showed no significant difference between parental and single AID-tagged cells in HCT116, yet the compound UHRF1-AID/DNMT1-AID line showed ~10% less DNA methylation than its wild-type counterpart (Fig. 1E). We also carried out LUMA in the DLD1 derivatives and found that the UHRF1-AID and DNMT1-AID cells had a small but significant reduction of DNA methylation (6% less than in the WT, Supplementary Fig. 1G).

Lastly, we compared the HCT116 WT and degron derivatives by RNA-seq, in the absence of auxin (Supplementary Fig. 1H). The tagging of UHRF1 or DNMT1 led to the deregulation of a very

small number of genes, the maximum being 6 genes downregulated in UHRF1-AID relative to WT. In contrast, the compound UHRF1-AID/DNMT1-AID line differed more markedly from WT, with 124 genes down, and 132 genes up (Supplementary Fig. 1H).

Collectively these results confirm that the tags added to UHRF1 and DNMT1 individually do not measurably affect cell viability, growth, or nuclear localization, and have minimal effects on DNA methylation and gene expression, therefore validating their use for functional analyses.

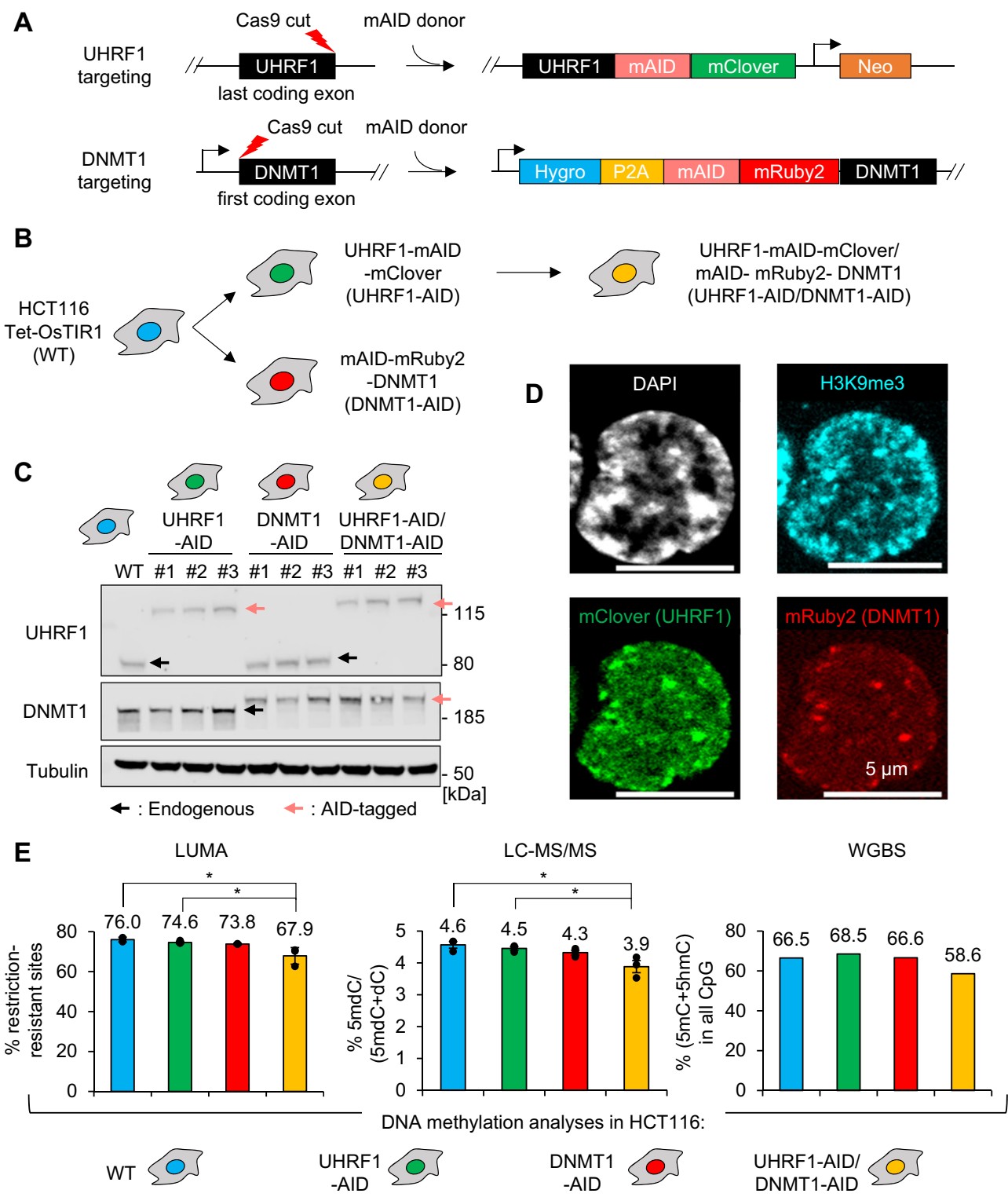

Fig. 1 | **Establishment and validation of endogenous AID-tagged UHRF1 and/or DNMT1 HCT116 cells. A** Schematic of the CRISPR/Cas9 genome editing strategy to endogenously tag UHRF1 with mAID/mClover and DNMT1 with mAID/mRuby2. **B** Order of events for the generation of the different cell lines. **C** Immunoblot images for validation of endogenous AID-tagged UHRF1 and/or DNMT1 HCT116 cells. Experiments in each panel were performed at least three times, and the representative results are shown. **D** Representative fluorescence images on UHRF1-AID/DNMT1-AID HCT116 cells showing that tagged UHRF1 and DNMT1 co-localize. **E** Quantification of the DNA methylation level in each HCT116 cell line with LUMA, LC-MS/MS, or WGBS. The p value is calculated with one-way ANOVA and Tukey's HSD test (*p < 0.05). Data are presented as mean values +/− SEM from biological triplicates. Source data are provided as a Source Data file.

## The depletion of UHRF1 and/or DNMT1 is efficient and causes growth arrest

After validating these basal conditions, we next tested the effects of triggering the degradation of UHRF1 and/or DNMT1 in the AID-tagged cell lines. Western blotting revealed that, as early as 2 h after treatment with auxin, UHRF1 and/or DNMT1 protein levels in HCT116 and DLD1 cells became undetectable, and that this depletion persisted as long as auxin was present (Fig. 2A and Supplementary Fig. 2A). We have noted in 3 independent clones that the degradation of DNMT1 and UHRF1 in the compound mutant cells is equally rapid but incomplete by ~8 h after treatment with auxin (Fig. 2A); this could possibly reflect saturation of the degradation system[31].

We then measured cell proliferation after auxin addition, using Incucyte videomicroscopy. The control cells (expressing OsTIR1 but having no AID-tagged protein) grew vigorously in the presence of auxin, as expected. However, cells depleted for UHRF1 and/or DNMT1 proliferated significantly slower than the control cells (Fig. 2B). This decrease in cell proliferation was markedly more pronounced after UHRF1 depletion than after DNMT1 depletion, and the compound UHRF1/DNMT1 depletion had a slightly stronger effect than the single UHRF1 depletion (Fig. 2B). Incucyte measurements detect confluency, which depends not only on the number of cells but on their size as well, so we also performed standard cell counting; these data confirmed the slower proliferation in UHRF1-depleted compared to DNMT1-depleted HCT116 cells (Fig. 2C). A similar trend was seen in DLD1 cells, where UHRF1 depletion led to a stronger inhibition of proliferation than DNMT1 depletion (Supplementary Fig. 2B).

A previous study has reported that inducible DNMT1-KO in HCT116 cells caused mitotic catastrophe and apoptosis within 4 days[32], so we sought to determine whether the decrease in cell proliferation may result from cell death. For this, we measured cell viability with trypan blue staining every four days after auxin treatment, but we did not detect any significant cell viability loss (Supplementary Fig. 2C); instead there was a buildup of cells in the G1 fraction of the cell cycle (Supplementary Fig. 2D). We complemented these data with an RNA-seq analysis on cells before or after a 4-day auxin treatment. The treatment caused the significant deregulation of 1259 genes in UHRF1-AID cells, 402 genes in DNMT1-AID cells, and 2107 genes in UHRF1-AID/DNMT1-AID cells (Supplementary Fig. 2E); in all cases, there were 5- to 10-fold more upregulated genes than downregulated genes. Seventy-four percent of the genes upregulated upon DNMT1 depletion were also induced by UHRF1 depletion (Supplementary Fig. 2F), but conversely, 77% of the genes upregulated by UHRF1 depletion were not induced upon DNMT1 depletion. This proves that the UHRF1 and DNMT1 have common target genes, as expected, but that UHRF1 depletion has broader and/or faster consequences.

As expected, the cancer/testis genes, which are repressed by DNA methylation[33], were among the upregulated genes (Supplementary Fig. 2G). In contrast, genes associated with cellular proliferation, in particular E2F targets, were among the downregulated genes in UHRF1-depleted and DNMT1-depleted cells (Supplementary Fig. 2G), in line with their slower growth and accumulation in G1.

Together these results indicate: that UHRF1 and/or DNMT1 depletion occurs effectively in the AID-tagged cell lines; that this depletion leads to profound growth retardation without detectable cell death; and that UHRF1 depletion has a more severe effect than DNMT1 depletion.

## Genetic rescues identify the domains of UHRF1 and DNMT1 critical for supporting growth

We next investigated the mechanism underlying the growth retardation. For this, we used genetic rescue of the AID-tagged HCT116 cell lines with DNMT1 and UHRF1 variants bearing point mutations in their critical domains (Fig. 2D). All the mutant proteins were expressed at levels similar to, or slightly higher than, the corresponding endogenous protein (Supplementary Fig. 2H, I).

For the UHRF1 rescue constructs, we observed that the exogenously expressed WT and TTD mutant rescued cell proliferation to a similar extent (Fig. 2E). In contrast, inactivating the UBL, PHD, SRA, or RING domain rendered UHRF1 non-functional for supporting growth (Fig. 2E).

The WT DNMT1 construct and its PBD mutant derivatives both rescued the cell proliferation (Fig. 2F). In contrast, the UIM mutant, H3K9me3 binding motif mutant, or catalytically inactive form of DNMT1 were all unable to rescue the slow growth phenotype.

To summarize, some but not all of the domains of UHRF1 and DNMT1 are required to support cell proliferation in HCT116 cells. The links between the proliferation defect and DNA methylation loss are explored in the following sections.

## UHRF1 depletion induces a more severe DNA methylation loss than DNMT1 depletion

We then examined the dynamics of DNA methylation loss upon removal of UHRF1 and/or DNMT1. As above, we started our experiments with the HCT116 cells and used 3 independent methods that measure DNA methylation levels: LUMA, LC-MS/MS, and shallow-coverage WGBS.

LUMA showed that the parental cells (WT) displayed no change in DNA methylation over the course of a 12-day auxin treatment (Fig. 3A). In contrast, cells depleted of UHRF1 and/or DNMT1 progressively lost DNA methylation, as expected (Fig. 3A). Strikingly, UHRF1 depletion caused a markedly stronger loss than DNMT1 depletion; for instance, 6 days after treatment, the percentage of restriction-resistant sites was ~75% in WT cells, ~55% in DNMT1-depleted cells, and ~40% in UHRF1-depleted cells (Fig. 3A). The cells depleted for both UHRF1 and DNMT1 had a slightly stronger loss than the cells lacking UHRF1 only. As a comparison, we included in the dataset "DKO" cells that have a DNMT3b deletion and a hypomorphic DNMT1 mutation[34], along with their control WT HCT116 cells. We observed that prolonged DNMT1 depletion did not elicit the deep loss of DNA methylation seen in DKO cells, whereas depletion of UHRF1, or DNMT1 and UHRF1, did have a comparable effect to the DKO mutation (Fig. 3A).

LC-MS/MS and WGBS results were fully consistent with the LUMA data (Fig. 3B, C, WGBS on individual clones in Supplementary Fig. 3A). In addition, LUMA on DLD1 degron cells showed that UHRF1 depletion caused a more severe loss of methylation than DNMT1 depletion in this cellular background as well (Supplementary Fig. 3B).

We also compared the magnitude of the effect caused by UHRF1 or DNMT1 depletion to that of a 5-azacytidine treatment (Fig. 3D). Removing UHRF1 for 4 days had the same effect as a 4-day treatment with 0.1 μM 5-aza (Fig. 3D). The 5-aza treatment decreased the abundance of DNMT1, but also that of DNMT3A and DNMT3B (Fig. 3E), which may explain why 5-aza caused deeper DNA demethylation than DNMT1 depletion.

Lastly, we used LUMA after auxin treatment to verify which of the rescue constructs can maintain DNA methylation levels following degradation of the endogenous UHRF1 or DNMT1 proteins (Supplementary Fig. 3C, D). The only mutant form of UHRF1 supporting DNA methylation maintenance was the TTD mutant (Supplementary Fig. 3C), while the only mutant form of DNMT1 that retained activity towards DNA methylation was the PBD mutant (Supplementary Fig. 3D). Therefore, for the 9 variants of UHRF1 and DNMT1 that we have tested, there is a one-to-one correspondence between the ability to support growth, and the ability to maintain DNA methylation.

Together these results further suggest that loss of DNA methylation underpins the growth retardation of the various degron lines treated with auxin. In addition, they show that UHRF1 depletion causes a more severe loss of DNA methylation than DNMT1 depletion, in parallel with a more severe growth retardation. Importantly, the slower

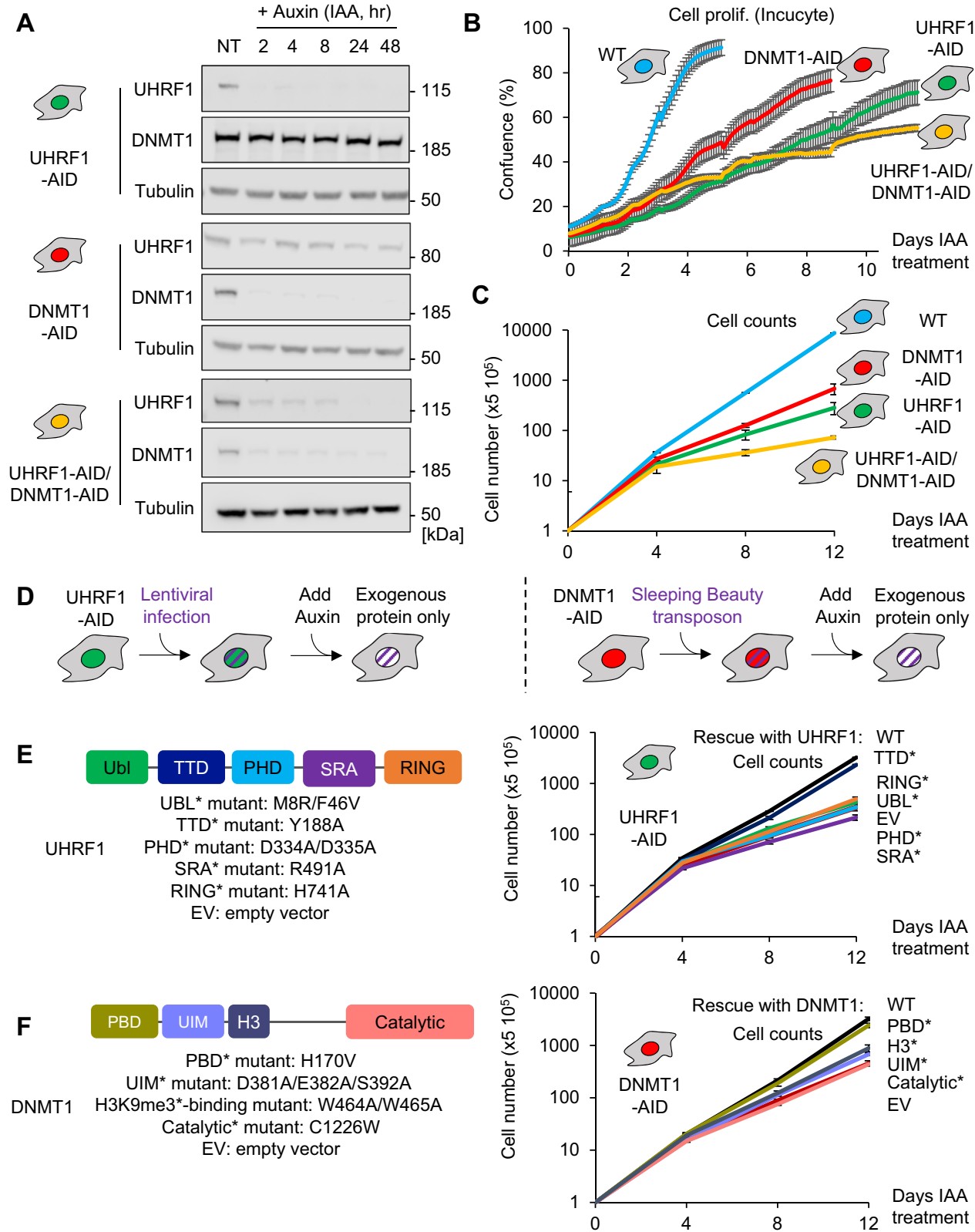

**Fig. 2 | The depletion of UHRF1 and DNMT1 is efficient, negatively affects growth, and can be rescued genetically. A** Immunoblot of HCT116 cells following treatment with Auxin (IAA) at the indicated time points (hours) and before treatment (NT). Experiments in each panel were performed at least three times, and the representative results are shown. **B** Cell proliferation of the HCT116 derivatives in the continuous presence of auxin for the indicated durations (Incucyte videomicroscopy). Error bars represent the SEM of biological triplicates. **C** Cell proliferation of the HCT116 derivatives in the continuous presence of auxin for the indicated durations (cell counting). The error bars represent the SEM of biological triplicates. **D** Schematic of the rescue experiments. **E** UHRF1 domain map showing the mutants studied (left panel) and corresponding cell proliferation analysis (Cell count, right panel). Error bars represent the SEM of three independent experiments. **F** Same as (**E**), but for DNMT1. Source data are provided as a Source data file.

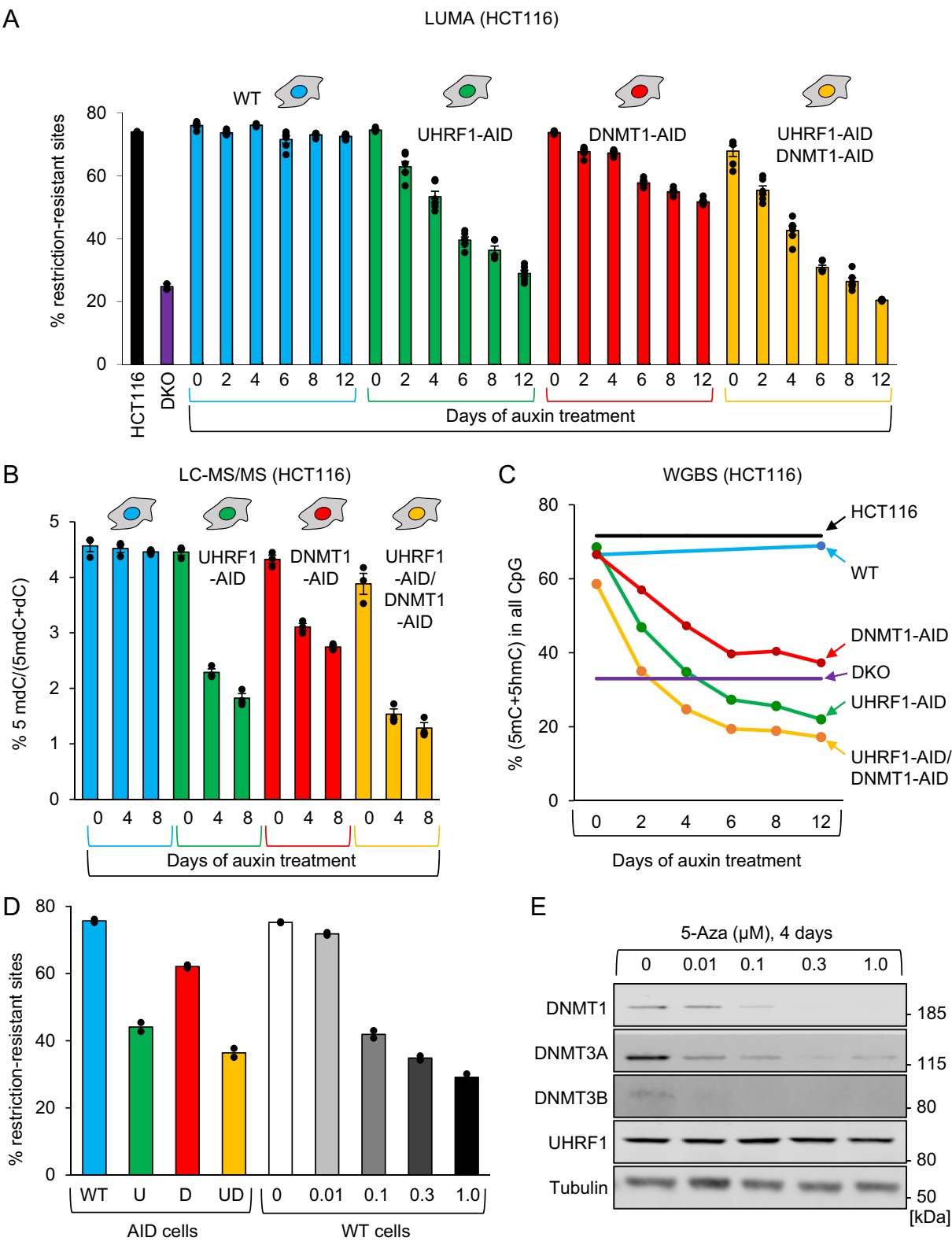

**Fig. 3 | UHRF1-depleted cells show more severe DNA hypomethylation than DNMT1-depleted cells. A** Global DNA methylation analysis in the indicated HCT116 derivatives after auxin treatment for the indicated duration (LUMA). Error bars represent the SEM of biological triplicates. **B** As in (**A**), but the quantitation of 5-mC was done by LC-MS/MS. **C** As in (**A**), but the quantitation of DNA methylation was done by WGBS. **D** LUMA analysis on cells treated with auxin or 5-aza. U: UHRF1-AID; D: DNMT1-AID; UD: UHRF1-AID/DNMT1-AID. **E** Western blotting on 5-Aza-treated cells. Experiments in each panel were performed two times, and the representative results are shown. Source data are provided as a Source data file.

growth of UHRF1-depleted cells rules out passive dilution of DNA methylation as an explanation for the greater loss of methylation they experience, when compared to DNMT1-depleted cells.

## UHRF1 depletion decreases DNA methylation at DNMT1, DNMT3A, and DNMT3B target sites

Our previous data clearly suggested that the role of UHRF1 in DNA methylation homeostasis goes beyond its canonical function of promoting DNMT1 activity. To get deeper insight into the mechanism(s) underlying this phenomenon, we performed deep-coverage WGBS, focusing on the early time points after auxin addition (days 0, 2, 4), which showed interesting dynamics yet minimize secondary effects due to growth differences.

For our analysis, we segmented the genome into 1-kb bins. Four days after auxin addition, cells lacking DNMT1 showed ~600,000 tiles that had lost 25% or more DNA methylation relative to day 0. However, that number was over twice as great in the UHRF1-depleted cells, which showed more than 1.3 million demethylated tiles (Fig. 4A). The joint depletion of UHRF1 and DNMT1 had an effect similar to, but slightly stronger than, UHRF1 depletion alone. A similar analysis performed only 2 days after auxin addition yielded similar results, albeit with smaller numbers of demethylated tiles (Supplementary Fig. 4A). The Venn diagrams of Fig. 4B and Supplementary Fig. 4B illustrate that most of the tiles demethylated after DNMT1 depletion were also demethylated after UHRF1 depletion.

We then refined this analysis by looking at distinct genomic regions (Supplementary Fig. 4C). The loss of DNA methylation in UHRF1 and/or DNMT1-depleted cells is pervasive and affects promoters, gene bodies, and intergenic regions. However, we noticed that gene bodies in particular experience greater loss of DNA methylation upon UHRF1 depletion than upon DNMT1 depletion (Supplementary Fig. 4C).

Gene-body methylation involves the de novo methyltransferases DNMT3A and DNMT3B[35–37], so the results prompted us to examine whether UHRF1 might have an effect on the targets of DNMT3A and DNMT3B, which are expressed in HCT116 cells.

In previous studies, kinetic DNA methylation studies performed with randomized oligonucleotides have determined systematically which flanking sequences are favored by DNMT1, DNMT3A, and DNMT3B in vitro, and the in vitro preferences are reflected in the cellular DNA methylation patterns[38–42]. We have exploited these data in the following manner (Fig. 4C): for each of the enzymes, we created a table in which the 256 possible NNCGNN sequences are ranked by order of preference in vitro. In parallel, we ranked the 256 possible NNCGNN sequences by average methylation level in each point of our WGBS dataset. Then we calculated pairwise Pearson $r$-correlation coefficients between the in vitro preferences and the actual WGBS values. This bioinformatic approach quantifies how much the flanking sequence preferences of a particular enzyme match to the actual genome-wide methylation in cells.

Figure 4C shows the results for 4 conditions: DNMT1-AID and UHRF1-AID cells, each before and 4 days after auxin addition. The data show that, before auxin is added, there is high correlation between the in vitro DNMT1 and DNMT3A preferences, and the actual average methylation levels in NNCGNN bins observed in cells, suggesting that these two enzymes have a strong contribution in shaping the methylome of HCT116 cells under our experimental conditions, which is not the case for DNMT3B (correlation score close to zero). When DNMT1 was depleted by auxin addition, its most preferred target sites were no longer among the most methylated ones, as the correlation coefficient dropped from 0.423 to 0.183. In contrast, the sites favored by DNMT3A were less affected, as the coefficient only marginally declined from 0.443 to 0.338. Therefore, DNMT1 depletion seems to affect preferentially DNMT1 target sites, as expected, providing a validation of our analysis.

After UHRF1 depletion, the preferred DNMT1 sites lost methylation as well, which was also expected. Notably, the drop was more profound after UHRF1 depletion (from 0.436 to −0.078) than after DNMT1 depletion. As DNMT1 is already completely depleted in the DNMT1-AID cells, this means that another activity contributing to methylation of the DNMT1 sites is also decreased in the UHRF1-AID cells. Interestingly, UHRF1 depletion also had a very strong effect on the DNMT3A sites, for which the correlation score went from 0.430 to 0.070, suggesting that the enzyme was no longer a major contributor to the DNA methylation pattern. The values for DNMT3B went from 0.040 to −0.451, indicative of a strong anticorrelation, and meaning that the best DNMT3 sites actually fell among the least methylated sites when UHRF1 was removed.

To summarize, this rich dataset shows that UHRF1 depletion leads to profound decreases of DNA methylation not just at the best DNMT1 target sites, but also at the best DNMT3A and DNMT3B target sites suggesting that UHRF1 also has a role in DNMT3A and DNMT3B mediated DNA methylation.

We obtained further support for this scenario by examining DNA methylation losses at H3K36me3-marked CpG islands, which are a well-described target of de novo methyltransferases in HCT116 cells[36]. We extracted from our WGBS data the methylation values for CpG islands and ranked them in 10 bins according to their H3K36me3 content (Fig. 4D). CpG islands with low levels of H3K36me3 lost the same amount of DNA methylation after UHRF1 depletion or after DNMT1 depletion: the methylation difference between these two conditions was close to zero. In contrast, CpG islands with higher levels of H3K36me3 lost significantly more methylation when UHRF1 was removed than when DNMT1 was removed (Fig. 4D). As a control, we carried out the same analysis with H3K79me2, another histone mark that is also found in gene bodies but is not associated with de novo DNA methyltransferases (Supplementary Fig. 4D). In that case we found no correlation between H3K79me2 levels and reliance on UHRF1. This analysis shows that regions of the genome that are especially reliant on de novo methyltransferases to gain DNA methylation are also especially reliant on UHRF1 to maintain their DNA methylation.

## Physical, functional, and genetic interactions between UHRF1 and the de novo methyltransferases

To test a possible physical association between UHRF1 and DNMT3A or DNMT3B, we performed a series of co-immunoprecipitation (co-IP) experiments. These experiments showed that UHRF1 indeed interacts with both DNMT3A and DNMT3B (Fig. 5A); furthermore the TTD domain was sufficient for interaction (Fig. 5A). We repeated these co-IP with full-length UHRF1 in the presence of Ethidium Bromide and obtained identical results, indicating that the interactions are not bridged by chromatin (Supplementary Fig. 5A, B). In addition, we could also demonstrate interaction between the endogenous proteins (Supplementary Fig. 5C, D)

Work with UHRF1 deletion mutants showed that the TTD and PHD domain were necessary for interaction with DNMT3A and DNMT3B, whereas the UBL, SRA, and RING finger were not (Fig. 5B). As the experiments pointed to an important role of the TTD, we performed a last series of co-IP experiments, with a mutant form of UHRF1 that is full-length but has two mutations (Y188A/Y191A) that inactivate the hydrophobic pocket of the TTD. The mutations significantly reduced the capacity of UHRF1 to interact with both DNMT3A and DNMT3B (Fig. 5C). In the light of these results, it appears surprising that UHRF1 with a mutant TTD can fully rescue the absence of the wild-type protein (Supplementary Fig. 3C). We hypothesize that, quantitatively, the main function of UHRF1 is to activate DNMT1, which is TTD-independent. To unmask a possible DNMT1-independent contribution of the TTD, we took cells depleted of both DNMT1 and UHRF1, in which we re-expressed either WT or TTD-mutated UHRF1

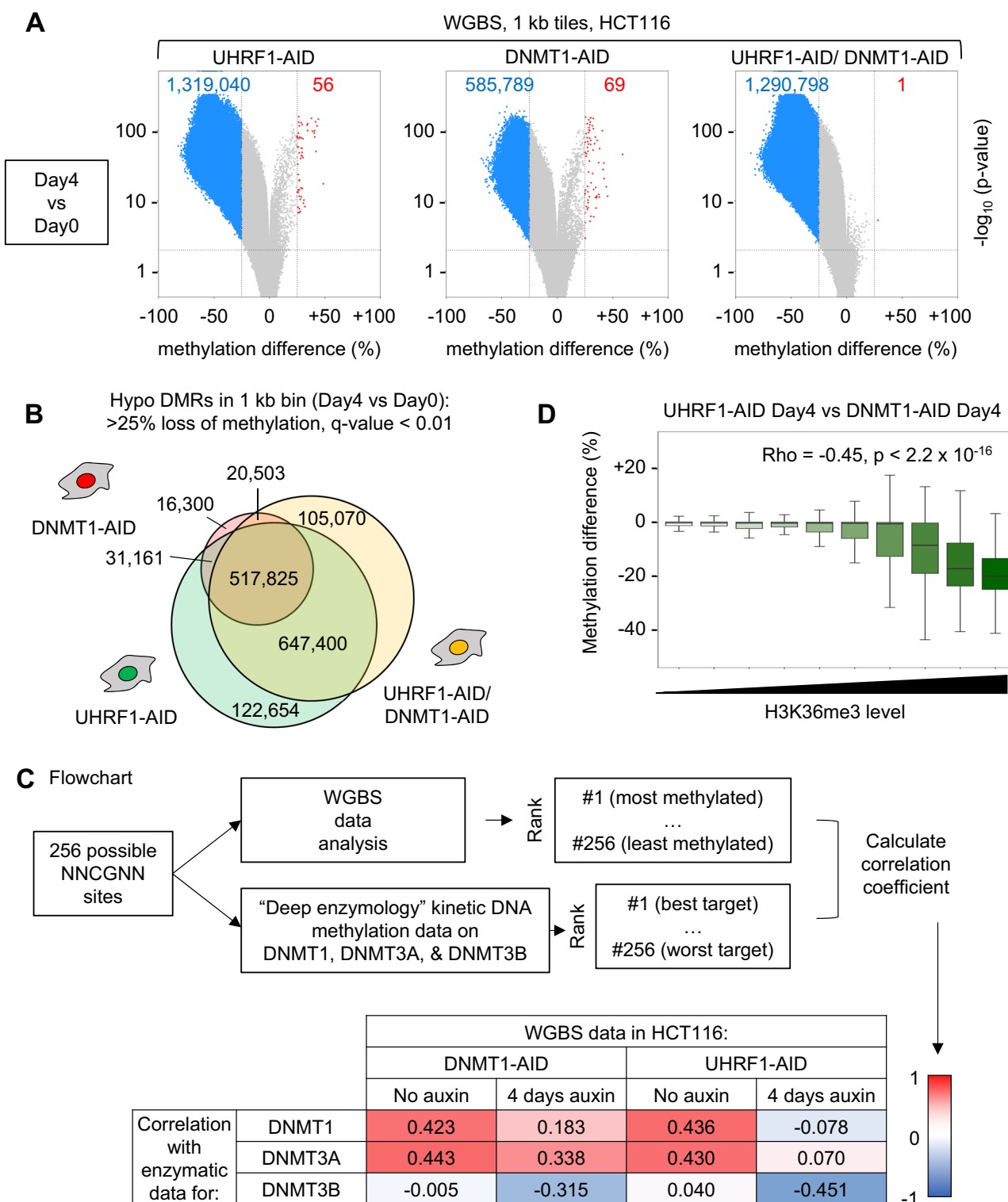

**Fig. 4 | Greater loss of DNA methylation upon UHRF1 depletion than upon DNMT1 depletion; UHRF1 regulates DNA methylation at DNMT1, DNMT3A and DNMT3B sites. A** Volcano plot of differentially methylated regions (DMRs, 1 kb bins) after 4 days of depletion of UHRF1 and/or DNMT1. Blue dots: hypomethylated regions (>25% loss of methylation, $q$-value < 0.01), red dots: hypermethylated regions (>25% gain of methylation, $q$-value < 0.01). Gray dots: no significant change. The $p$ value is corrected to $q$-value using sliding linear model (SLIM). **B** Venn diagram of the hypomethylated regions in the indicated cell lines, 4 days after depletion of the proteins. **C** Workflow used to quantitatively compare WGBS methylation values to the in vitro preferences of DNMT1, DNMT3A and DNMT3B. **D** Higher levels of H3K36me3 correlate with larger losses of DNA methylation in CpG islands. The CGIs were ranked by H3K36me3 level in HCT116 cells and split into 10 equally sized bins. DNA methylation differences were estimated from biological triplicates. Lines = median; box = 25th–75th percentile; whiskers = 1.5 × interquartile range from box. The $p$ value is from Spearman's correlation tests. Source data are provided as a Source data file.

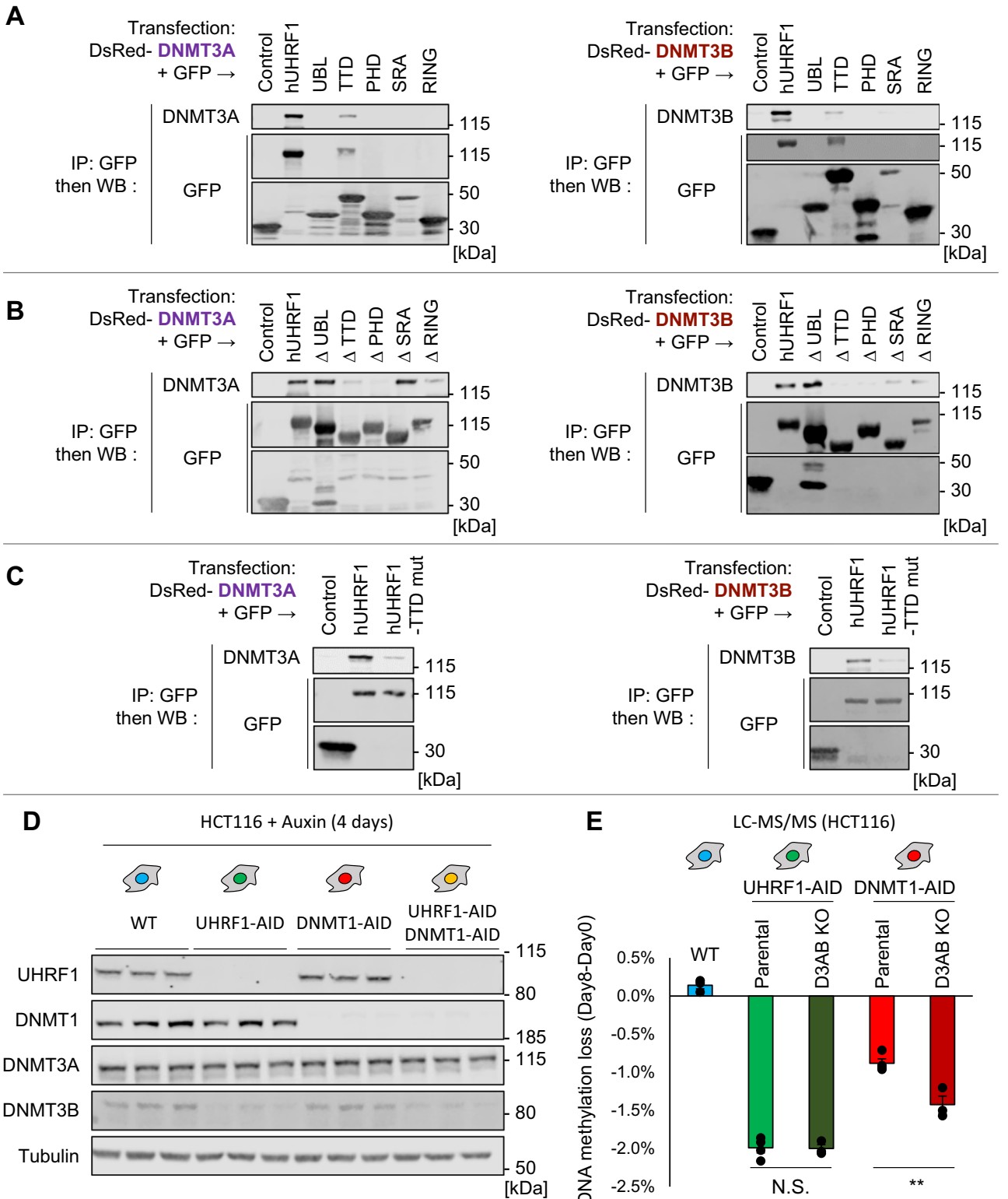

**Fig. 5 | Physical and functional interaction between UHRF1, DNMT3A, and DNMT3B. A** Western blotting after the indicated co-immunoprecipitation experiments. hUHRF1: Full-length protein. The other names indicate isolated domains, as depicted in Fig. 2E. **B** Same as in (**A**), except we used truncated constructs in which the indicated domains were deleted from the full-length protein. **C** Same as in (**A**), except we used a full-length UHRF1 protein with a point mutation in the Tandem Tudor Domain (Y188A/Y191A). **D** Western blotting showing abundance of the indicated proteins in total cell extracts. **E** Quantitation of the loss of DNA methylation in the indicated cell lines after 8 days of protein depletion, by LC-MS/MS. The p value is calculated with one-way ANOVA and Tukey's HSD test (N.S. $p > 0.05$, **$p < 0.01$). Data are presented as mean values +/− SEM from biological triplicates. **A**−**D** Experiments in each panel were performed at least two times, and the representative results are shown. Source data are provided as a Source data file.

(Supplementary Fig. 5E). In this condition, the TTD-mutated version of UHRF1 was less efficient at rescuing than the WT allele.

To summarize, we detect a physical interaction between UHRF1 and the two de novo methyltransferases in HCT116 cells, this interaction involves the TTD, and it is not indirectly mediated by chromatin.

We identified a further mechanistic link between UHRF1 and the de novo methyltransferases by a fully independent approach. We recently developed a highly sensitive proteomic approach to characterize the "chromatome" of cells in culture, i.e. to isolate and quantify by mass spectrometry the proteins associated with chromatin[43]. We applied this technique of our cell lines at various time points after DNMT1 or UHRF1 depletion, and observed that one of the proteins that was less abundant in chromatin after UHRF1 removal than after DNMT1 removal was DNMT3B (Supplementary Fig. 5F). We carried out western blotting on whole-cell lysates and found that UHRF1 depletion had no discernible effect on the amount of DNMT1 or DNMT3A, but that it led to a decrease of DNMT3B abundance, while DNMT1 depletion had no such effect (Fig. 5D). The decrease of DNMT3B on chromatin in the absence of UHRF1 is therefore mirrored in whole-cell extracts.

We then explored the genetic interactions between UHRF1, DNMT1, DNMT3A, and DNMT3B. For this, we generated CRISPR knockouts of DNMT3A and DNMT3B in the DNMT1-AID and UHRF1-AID lines (Supplementary Fig. 5G), and observed their effects on DNA methylation levels. As expected, removing DNMT3A and DNMT3B from the DNMT1-AID line (D3AB DKO derivative) led to a greater loss of DNA methylation upon auxin treatment (Fig. 5E). In contrast, the D3AB DKO mutations did not make the loss of methylation more severe in the UHRF1-AID line (Fig. 5E). This important result suggests that UHRF1 does not act in parallel to, but instead upstream of, DNMT3A and DNMT3B, which is consistent with our co-IP results.

Lastly, these genetic experiments brought another crucial conclusion: the DNMT1-AID/DNMT3A KO/DNMT3B KO, which are completely devoid of DNMT activity upon auxin addition, still lose DNA methylation more slowly than the UHRF1-AID line treated with auxin (Fig. 5E). Therefore, besides stimulating the activity of the DNA methyltransferases, UHRF1 must be preserving DNA methylation homeostasis by at least one other mechanism.

### UHRF1 opposes active demethylation by TET2

To guide the next set of experiments, we went back to our WGBS data. The sequence preferences of TET1 and TET2 have been identified in vitro[44], and we asked whether the optimal target sites of these enzymes were particularly likely to lose methylation in the absence of DNMT1 or UHRF1. We used the same workflow described earlier in Fig. 4C, and calculated correlation coefficients between WGBS-derived methylation data and in vitro data for the TET enzymes (Fig. 6A).

We found that the optimal TET1 and TET2 sites became strongly hypomethylated upon DNMT1 removal (correlation coefficients of −0.330 and −0.451 respectively). However, the demethylation at these sites became even more marked after UHRF1 was removed (coefficients of −0.498 and −0.579 respectively). This result is compatible with heightened TET action upon UHRF1 removal, suggesting that UHRF1 might oppose TET activity.

We tested this possibility genetically, focusing on TET2, which is the more expressed enzyme in HCT116 cells. For, this, we generated stable shTET2 derivatives of our UHRF1-AID, DNMT1-AID, and UHRF1-AID/DNMT1-AID HCT116 lines. The knockdown efficiency was ~80% at the mRNA level (Fig. 6B). We then measured DNA methylation by the LUMA assay in the various shCtrl and shTET2 lines, before and after auxin addition.

In the absence of auxin treatment, shTET2 led to a small but significant increase of DNA methylation, only in the DNMT1-AID and UHRF1-AID/DNMT1-AID lines (Supplementary Fig. 6A). Upon 4 days of auxin treatment, the UHRF1-AID, DNMT1-AID, and compound UHRF1-AID/DNMT1-AID lines expressing non-targeting shRNA lost DNA

methylation to various extents, with the cells lacking UHRF1 losing more DNA methylation than the cells lacking DNMT1 (Fig. 6C), which agrees with all of our previously presented data.

We then examined the effects of shTET2 combined with auxin treatment. In the DNMT1-AID line, the shTET2 did not rescue the DNA methylation loss, suggesting that active demethylation by TET2 is not the main contributor in this situation. In contrast, the shTET2 did significantly alleviate the DNA methylation loss experienced by UHRF1-AID or UHRF1-AID/DNMT1-AID cells, as shown both by LUMA (Fig. 6C) and LC-MS/MS (Fig. 6D). This key result establishes that TET2 activity contributes to DNA methylation loss when UHRF1 is absent, but not when DNMT1 is absent. Similar results were obtained after 8 days of auxin depletion (Supplementary Fig. 6B). In addition, we measured cell proliferation in all the cell lines to eliminate possible confounding factors (Supplementary Fig. 6C). In all cases, the shTET2 derivatives grew faster than the matched shCtrl line. Therefore, shTET2 does not preserve DNA methylation in UHRF1-depleted cells by preventing passive DNA methylation.

We therefore conclude that UHRF1 protects the genome against TET2 activity, which contributes to the more severe DNA hypomethylation seen in UHRF1-depleted cells, as compared to DNMT1-depleted cells, or even cells lacking all three DNMTs (See model in Fig. 7).

## Discussion

Using degron tools, we have carried out a precise time-resolved analysis of DNA methylation loss upon removal of UHRF1, DNMT1, or both. Our genomics data coupled to genetic experiments show that, in addition to its well-described role as an activator of DNMT1, UHRF1 also interacts functionally with DNMT3A and DNMT3B. In addition, we show that UHRF1 opposes the DNA demethylating activity of TET2. Besides their conceptual importance, these findings may be relevant for developing future cancer therapies.

### A powerful tool to study UHRF1 and DNMT1 function

We generated colorectal cancer cell lines in which the endogenous copies of UHRF1 and/or DNMT1 are tagged with fluorescent markers as well as degron tags, allowing for their rapid and controlled depletion. These cell lines constitute a valuable resource for research into the dynamics and functions of these two essential epigenetic regulators.

In the absence of auxin, the fluorescently labeled UHRF1 and DNMT1 proteins appear fully functional (Fig. 1D and Supplementary Movies). In addition, we chose the mClover/mRuby fluorescent protein pair because it can be used for FRET analysis[45]. This provides an ideal system with which to study the abundance, localization and dynamics of these two key epigenetic actors further in the future.

The proteins are rapidly, fully, and synchronously degraded upon auxin addition, allowing us to examine DNA demethylation dynamics upon removal of the key regulators. This question has been addressed in the past, by using shRNA[24], by transfecting the Cre protein into conditional KO cells[46], or by high-density CRISPR scanning[47]. However our degron approach has unprecedented temporal resolution and population homogeneity, permitting more precise analyses. One limitation of the degron approach, though, is the addition of a tag on the endogenous proteins, which could possibly alter their function even in the absence of auxin. We chose the location of the tag (Nter vs. Cter) with this problem in mind, nevertheless it remains that the doubly tagged lines have a small yet significant defect of DNA methylation. Steps that could be examined in the future to solve this issue include the use of smaller tags.

### Roles of UHRF1 and DNMT1 in cancer cell proliferation or viability

The addition of auxin to our AID-tagged cells leads to a rapid and extensive decrease in UHRF1 and/or DNMT1 protein abundance. This

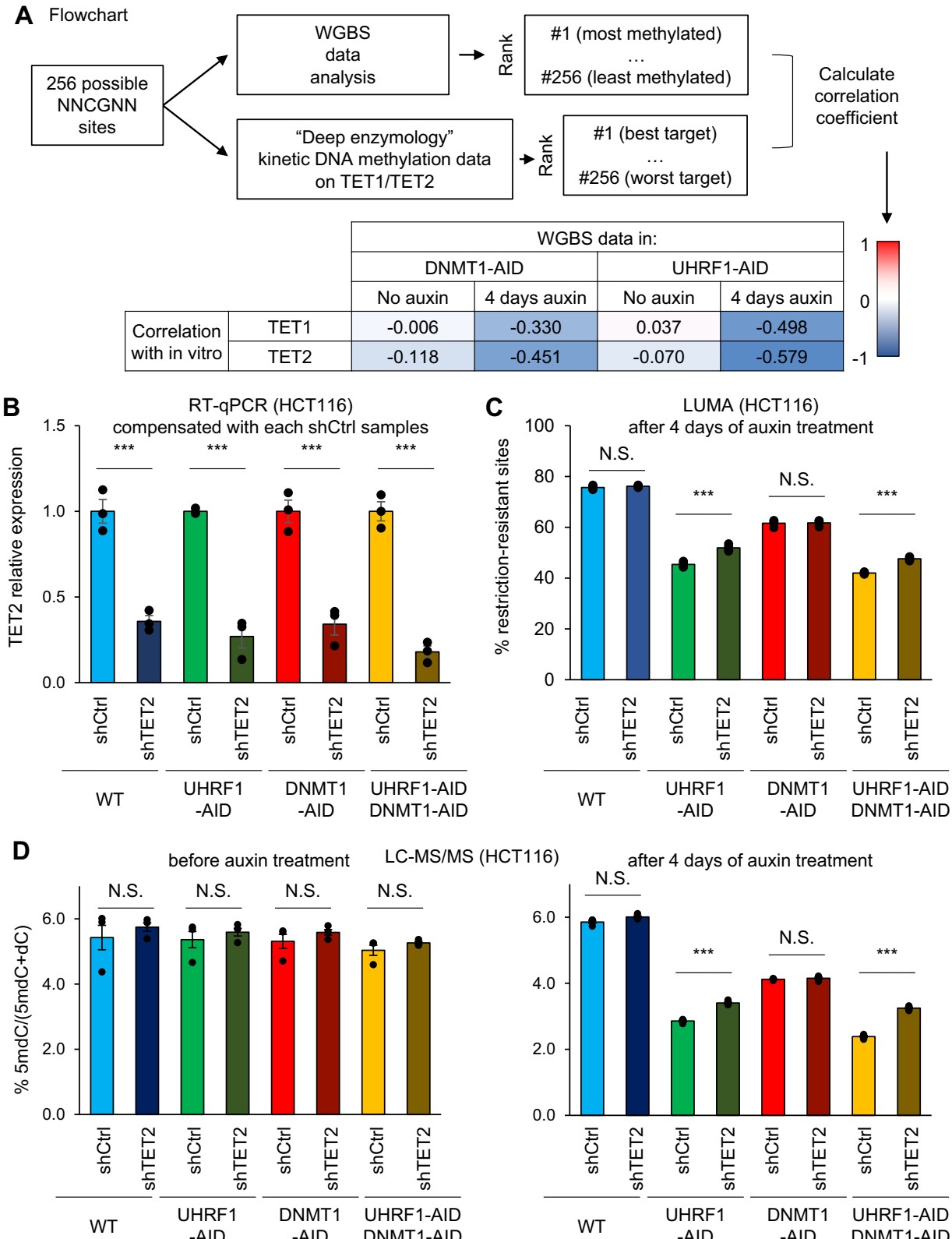

**Fig. 6 | UHRF1 protects against active demethylation by TET2. A** Workflow used to quantitatively compare WGBS methylation values to the in vitro preferences of TET1 and TET2. **B** RT-qPCR analysis for validation of TET2 knockdown for HCT116 UHRF1 and/or DNMT1-AID cell lines. **C** Global DNA methylation analysis (LUMA) for HCT116 UHRF1 and/or DNMT1-AID cell lines combined with TET2 knockdown. **D** LC-MS/MS analysis on the indicated samples. Error bars represent the SEM of three independent experiments. The *p* value is calculated using two-sided Student's *t* test (N.S. *p* > 0.05, ***p < 0.001). Source data are provided as a Source data file.

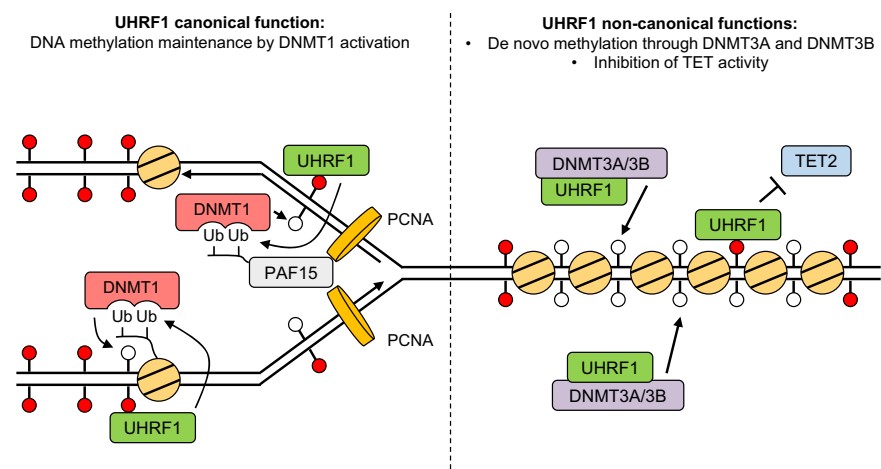

**Fig. 7 | A revised and expanded model for UHRF1 functions in DNA methylation homeostasis.** Left panel: the canonical function of UHRF1 is to promote DNA methylation maintenance by DNMT1. Right panel: we demonstrate non-canonical functions of UHRF1 that are independent of DNMT1 but contribute to DNA methylation homeostasis.

leads to a severe impairment of cell growth both in HCT116 and DLD-1 cells, yet the cells maintain viability.

These results are consistent with a recent report describing DNMT1-degron cells[30], yet they contrast with earlier publications: most notably, the inducible deletion of the DNMT1 gene in HCT116 cells has been reported to cause a G2 arrest, eventually followed by escape and mitotic catastrophe[32]. Possible causes for this discrepancy with our observations might include the removal of an uncharacterized important genetic element along with the targeted DNMT1 genomic sequence and/or the expression at low levels of a truncated DNMT1 protein that has negative consequences in the knockout cells. However, we cannot rule out the possibility that minute amounts of DNMT1 escaping degradation in our system are sufficient to promote survival.

Similarly, previous reports in which UHRF1 was depleted by siRNA or shRNA reached various conclusions as to the effects of the depletion[24,26]. Removal of the protein by a CRISPR KO has been attempted, but only yielded hypomorphs[48], suggesting that the protein might be essential. In our study, we observed a strong cell proliferation defect after UHRF1 depletion compared with WT cells (Fig. 2B, C). This likely explains why UHRF1 KO have not yet been reported in cancer cells. It also suggests that caution should be exercised when carrying out and interpreting siRNA or shRNA experiments on UHRF1, as the least depleted cells will have a growth advantage over the most depleted ones.

The mechanisms underpinning the essentiality of UHRF1 and DNMT1 for long-term cancer cell proliferation have been suggested to be linked with their role in DNA methylation homeostasis[24]. Our rescue experiments are compatible with this hypothesis, as mutants that rescue DNA methylation also rescue growth, and vice versa. However, the number of mutants we and others have examined is still limited, and the mutations studied, such as the RING finger inactivation, may affect other important functions in addition to DNA methylation maintenance. The tools we have developed may help reveal if the functions of DNMT1 and UHRF1 in cell proliferation and DNA methylation maintenance are indeed fully linked, or whether they can be dissociated.

### Functional and physical interaction between UHRF1, DNMT3A and DNMT3B

There have been some indications in the past that UHRF1 might be connected to the de novo DNA methylation machinery[49–51] but our results now rigorously establish this connection, ground it in molecular detail, and determine its effects on DNA methylation genome-wide.

The physical interaction between proteins involves the TTD of UHRF1 and, more precisely still, its hydrophobic pocket. Our co-immunoprecipitations in the presence of Ethidium Bromide eliminate the possibility that the interaction is bridged by chromatin, however we cannot presently conclude whether the interaction is direct, or involves other unknown factors. We note that DNMT3A contains a histone-like TARK motif that is methylated on the lysine by G9A and GLP[52]. This situation is reminiscent of other proteins directly bound by the TTD, namely histone H3 and DNA Ligase 1[10,11]. Thus, one possibility for future exploration will be to test the possibility that UHRF1 interacts directly with the TARK motif of DNMT3A.

We find that depleting UHRF1 leads to decreased abundance of the DNMT3B protein, without affecting DNMT1 or DNMT3A (Fig. 5D). Additional experiments could be carried out in the future to identify the underlying mechanism which could be direct or indirect, for example depending on the fact that methylated nucleosomes appear to stabilize DNMT3B[53].

Lastly, we have carried out our experiments in human cancer cells, but it will be worthwhile in the future to clarify whether UHRF1 also promotes DNMT3A/DNMT3B activity in other systems, such as mouse embryonic stem cells.

### UHRF1 inhibits TET2 activity

Our epistasis studies reveal that TET2 contributes to DNA demethylation more actively when UHRF1 is absent. This finding may at first sight appear discordant with a recent report, which found that UHRF1 actually recruits the short form of TET1 to heterochromatin, where it catalyzes DNA hydroxymethylation[54]. However, disparities in cellular systems, coupled to dissimilarities between TET1 and TET2, could contribute to the contrast between our results. Also, we note that the recruitment of TET1 by UHRF1 appears to be limited to the late S-phase, and could be counterbalanced by other processes in other phases of the cell cycle.

At this stage, we cannot say if the decreased TET2 action is due to an inhibition at the level of transcription, translation, stability, or activity of the protein. However, an interesting parallel might possibly be drawn with results obtained in mouse ES cells, where UHRF1 has been proposed to inhibit SETDB1 activity by binding hemimethylated DNA[55]. A similar regulation might occur between UHRF1 and the TETs.

### UHRF1 as a therapeutic target in cancer

Cancer cells have an aberrant epigenome, and this creates opportunities for anti-tumoral therapies[56]. Among the various epigenetic marks,

DNA methylation has been validated as a valuable target[21]. The DNMT1 inhibitor 5-aza-cytidine is successfully used in the clinic against Myelodysplasia and Acute Myeloid Leukemia but has limitations such as high toxicity, rapid degradation, and emergence of resistance[57]. The new generation of selective DNMT1 inhibitors that has been developed[58] may alleviate some of those issues, yet these molecules still trigger DNMT1 degradation[59], which might have unwanted side effects. Our data point out that an altogether different strategy may be viable, by targeting UHRF1 instead of DNMT1, which justifies drug design efforts currently ongoing in the community[60–63]. As with any essential protein, one of the challenges will be to identify a therapeutic dosage window and/or appropriate delivery methods such that cancer cells are harmed while healthy cells are spared. It is possible that the high expression levels of UHRF1 in tumors[23,24] will provide such a window. Altogether, our work reveals non-canonical functions of UHRF1, and open up avenues for further exploration of this key epigenetic regulator in normal cells and in disease.

## Methods

### Plasmid construction

We utilized the pX330-U6-Chimeric_BB-CBh-hSpCas9 plasmid, obtained from Feng Zhang (Addgene #42230), as the basis for constructing CRISPR/Cas vectors. To generate the mAID donor plasmids, we modified constructs of the Kanemaki lab (Addgene #72827 and #121180). In order to incorporate mRuby2, we replaced mCherry2 in the donor plasmid (Addgene #121180).

For the rescue experiments, wild-type (WT) UHRF1 and each of the point mutants (M8R/F46V, Y188A, DAEA, G448D, and H741A) were cloned into pLenti6.2/V5-DEST (invitrogen). Likewise, WT DNMT1 and each of the point mutants (H170V, D381A/E382A/S392A, W464A/W465A, C1226W) were cloned into pSBbi-Bla (Addgene: #60526). To target DNMT3A and DNMT3B, we cloned the oligonucleotide sequences for gRNA into the lentiCRISPR v2-Blast vector (Addgene #83480). Additionally, we cloned the shRNA targeting TET2 into the pLKO.1-blast vector (Addgene #26655). Plasmids were generated using PCR, restriction enzymes, or Gibson Assembly Cloning techniques. All plasmids underwent sequencing prior to their utilization. The oligonucleotide sequences inserted into the LentiCRISPR v2-Blast vector and pLKO.1-blast vector are available in Supplementary Table 1.

### Cell culture, transfection, and colony isolation

The HCT116 cell line, which conditionally expresses OsTIR1 under the control of a tetracycline (Tet)-inducible promoter, was obtained from the RIKEN BRC Cell Bank (http://cell.brc.riken.jp/en/) and genotyped by Eurofins. HCT116 cell lines were cultured in McCoy's 5A medium (Sigma-Aldrich), supplemented with 10% FBS (Gibco), 2 mM L-glutamine, 100 U/mL penicillin, and 100 μg/mL streptomycin. The DLD1 cell line, which constitutively expresses OsTIR1 (F74A), was provided by the Kanemaki Lab. DLD1 cell lines were cultured in RPMI-1640 medium (Sigma-Aldrich), supplemented with 10% FBS (Gibco), 2 mM L-glutamine, 100 U/mL penicillin, and 100 μg/mL streptomycin. Both cell lines were maintained in a 37 °C humid incubator with 5% $CO_2$. The authentication for both cell lines was performed with STR profiling.

To establish stable cell lines, cells were seeded in a 24-well plate and transfected with CRISPR/Cas and donor plasmids using Lipofectamine 2000 (Thermo Fisher Scientific). Two days post-transfection, cells were transferred and diluted into 10-cm dishes, followed by selection in the presence of 700 μg/mL G418 or 100 μg/mL Hygromycin B. After a period of 10–12 days, colonies were individually picked for further selection in a 96-well plate.

For the induction of AID-fused protein degradation in HCT116 cell lines, cells were seeded and incubated with 2.0 μg/mL doxycycline (Dox) and 20 μM auxinole for 1 day. Subsequently, the medium was replaced with fresh medium containing 2.0 μg/mL Dox and 500 μM indole-3-acetic acid (IAA), a natural auxin. Similarly, to induce AID-fused protein degradation in DLD1 cell lines, cells were seeded and incubated with regular medium for one day, followed by medium replacement with 1 μM 5-Ph-IAA.

### Immunofluorescence staining

For the immunofluorescence staining experiments, 200,000 cells were seeded into one well per cell line per staining on an 18 mm:18 mm microscope slide of a 6-well coated 1 day before with Geltrex Ready-to-Use (Thermo Fisher Scientific). Two days after seeding, the cells were washed twice with PBS, fixed for 10 min in 4% PFA, and permeabilized for 5 min in Triton-X 100 and washed twice in between every step with 0,01% PBS-Tween 20. Primary antibody incubation of H3K9me3 was done in blocking solution (4% BSA) and was incubated overnight at 4 °C (rabbit polyclonal a-H3K9me3 antibody Abcam cat. 8898, 1:1000 dilution) followed the next day with the secondary antibody (donkey anti-mouse 647 antibody, 1:2000 dilution) in 37 °C for 1 h in a humid chamber. All the acquisitions were conducted with a Nikon TiE microscope equipped with a Yokogawa CSU-W1 spinning disk confocal unit (pinhole size 50 μm), together with an Andor Borealis illumination unit, [Andor ALC600 laser combiner (405 nm/488 nm/561 nm/640 nm)]. The images were acquired with an Andor IXON 888 Ultra EMCCD camera, with ×100/1.45NA oil immersion objective through the interface of the software NIS Elements (Version 5.02.00) in Perfect Focus System with lasers at 405 nm for Dapi counterstain, 488 nm for mClover, 561 nm for mRuby2 and 640 nm for H3K9me3. Images were acquired with the same laser power, acquisition time and gain.

### Videomicroscopy analysis

For live cell imaging, cells were grown on 35 mm FluoroDish (World Precision Instruments) with 0.17 mm thick optical quality glass bottom and fitted with a 4-well silicone insert (Ibidi). Timelapse images were taken every 10 min for 20 h using an inverted Eclipse Ti-E microscope (Nikon) equipped with a CSU-X1 (Yokogawa) spinning disk integrated in Metamorph software, and a 4-laser bench (Gataca systems). ~45 μm Z stacks were acquired (Z-step size: 3 μm) with a ×60 CFI Plan Apo VC oil-immersion objective (numerical aperture 1.4). The microscope has a motorized Nano z100 piezo stage (Mad City Lab), a stage top incubator (Tokai Hit) and an EMCCD camera (Evolve, Photometrics). The images were 3D deconvolved using the NIS Elements software (Nikon).

### Infection/transfection for rescue experiments

The generation of lentiviral or Sleeping Beauty transposon vectors followed the methodology of "Plasmid Construction." Subsequently, the cell lines were either infected or transfected with WT, UHRF1-AID, DNMT1-AID, or UHRF1/DNMT1-AID. To ensure stable expression of the target genes or shRNA, the infected or transfected cells were incubated with 10 μg/mL Blasticidin for a period of 1 week, allowing for the selection of stable cell populations.

### Western blot analysis

Cells were harvested after trypsinization, washed twice with PBS, and lysed with RIPA buffer (Sigma-Aldrich) with protease inhibitor (1 mM phenylmethanesulfonyl fluoride and 1× Complete Protease Inhibitor Cocktail; Roche), then sonicated with a Bioruptor (Diagenode). The sonicated samples were centrifuged at $16,000 \times g$ for 15 min, then the supernatants were subjected to the Bradford Protein Assay Kit (BioRad). Equivalent amounts of protein were resolved by SDS-PAGE and then transferred to a nitrocellulose membrane. Immunoblotting was performed following blocking with 5% skim milk in PBST using antibodies against UHRF1 (Santa Cruz Biotechnology, sc-98817, 1:1000), DNMT1 (Cell Signaling Technology, #5032, 1:1000), Tubulin (Abcam, ab7291, 1:5000), DNMT3A (Abcam, ab188470, 1:1000), DNMT3B (Cell Signaling Technology, 67259T, 1:1000), and GFP (Roche, 11814460001, 1:1000).

## Cell proliferation assay

For cell proliferation studies, HCT116 cells were seeded at a density of 5000 cells per well in a 96-well plate. They were then treated with 2.0 μg/mL Dox and 20 μM auxinole for 1 day. Following this, the medium was replaced with fresh medium containing 2.0 μg/mL Dox and 500 μM IAA. Throughout the experiment, images were captured every 2 h using an IncuCyte ZOOM microscope (Essen Bioscience). The IncuCyte ZOOM software was utilized to determine the cell confluency (%) based on the acquired images.

To obtain cell count data and assess cell viability, trypan blue staining was performed after every 4 days of auxin treatment. The TC20 Automated Cell Counter (BioRad) was used to obtain the cell count data and calculate the cell viability rate.

## Cell cycle analysis by flow cytometry with BrdU staining

WT, UHRF1-AID, DNMT1-AID, and UHRF1/DNMT1-AID cells were incubated with 10 μM 5-bromo-2′-deoxyuridine (BrdU) for 45 min at 37 °C in a $CO_2$ incubator. The cells were trypsinized and collected in a 15 mL tube. After washing with PBS, the cell pellet was resuspended in 750 μL PBS, then 2250 μL ice-cold ethanol added to fix the cells (3 mL final volume of 75% ethanol). Fixed cells were incubated for at least 30 min at −20 °C, and stored before performing flow cytometry analysis. Cellular DNA was denatured in 2 N HCl for 15 min, followed by pelleting the fixed cells and washing with PBS + 1% BSA. BrdU was then detected with the mouse anti-BrdU-FITC antibody (BD Biosciences) in PBS + 1% BSA. For cell cycle analysis, cells were rinsed with PBS + 1% BSA, then resuspended in PBS containing propidium iodide (1:500, Invitrogen) and 150 μg/mL RNaseA. Cells were incubated overnight at 4 °C in the dark. The percentages of cells in subG1, G1, S and G2/M phases were measured with FACSCalibur (BD Biosciences). Data were analyzed with the FlowJo software. The population was gated on size using SSC-A/FSC-A. To obtain single cell information, we also used FSC-H/ FSC-A gating. All gated information was supplied for cell population analysis.

## DNA methylation analysis

LUMA and Pyrosequencing analyses were conducted following standard procedures. Whole-genome bisulfite sequencing (WGBS) libraries were prepared using the tPBAT protocol, as described by Miura et al.[64,65]. The library preparation involved using 100 ng of genomic DNA spiked with 1% (w/w) of unmethylated lambda DNA from Promega. Subsequently, sequencing was carried out by Macrogen Japan Inc. utilizing the HiSeq X Ten system.

To process the sequenced reads, BMap was employed to map them to the hg38 reference genome as previously described[65]. A summary of the mapping information can be found in Supplementary Table 2. The methylation level of each cytosine was calculated only when it was covered by at least ten reads.

Once the methyl reports data was obtained, methylKit was utilized to determine the methylation levels of individual CpG sites and identify differential methylated regions (DMRs). In this analysis, DMRs were defined as having a methylation difference greater than 25% and a q-value lower than 0.01.

## Flanking sequence analysis

Genome-wide DNA methylation profiles were used to extract methylation level of individual CpG sites and their flanking sequences as described earlier[66]. CpGs with sequences coverage ≥10 were included in the downstream analysis. Enzymes' flanking sequence profiles were combined from published data[39–42,44]. Pearson r-values were determined with Microsoft Excel. Symmetrical preference profiles for DNMT3A and DNMT3B were generated by averaging the preferences of pairs of corresponding complementary flanks[38].

## RNA-seq library preparation and data processing

Total RNA was extracted from cells with RNeasy Plus Mini kit (Qiagen) according to the manufacturer's instructions and quantified using Qubit RNA BR Assay kit on Qubit 2.0 Fluorometer (Thermo Fisher Scientific). RNA quality was assessed by TapeStation (Agilent), requiring a minimal RNA integrity number (RIN) of 9.0. A total amount of 4 μg total RNA per sample was used as input material for the RNA sample preparations. Sequencing libraries were generated using NEBNext® Ultra™ RNA Library Prep Kit for Illumina (NEB), and high throughput sequencing was performed on Illumina NovaSeq 6000 platform using a 150 bp paired-end sequencing in Novogene Co., Ltd. Reads with low quality and adapter sequences were removed using Trimmomatic with default settings (version 0.38). Subsequently, the reads were aligned to the hg38 reference genome using STAR (v2.6.1d). FeatureCounts (v1.5.0-p3) was used to count the read numbers in each gene. Differential expression analysis was performed using the DESeq2 R package. The adjusted p value (FDR) < 0.05 and the absolute value of the log2 (fold change) >1 were used as the threshold to identify differentially expressed genes (DEGs).

## Gene set enrichment analysis (GSEA)

Gene set enrichment analysis was performed using GSEA (v4.1.0) and default parameters. "HALLMARK_E2F_TARGETS" gene sets was derived from the Molecular Signatures Database (MSigDB)[67]. Gene set for Cancer-Testis (CT) genes was obtained from Almeida et al.[68]. GSEA results with FDR < 0.25 and absolute value of Normalized Enrichment Score (NES) >1 were used for threshold to assess statistical significance[69].

## ChIP-seq analysis

ChIP-seq data for HCT116 cells was obtained from ENCODE. Upon downloading the data, we performed quality checks on the reads using FASTQC (v0.11.9, available at https://www.bioinformatics.babraham.ac.uk/projects/fastqc). Reads with low quality and adapter sequences were removed using Trimmomatic with default settings (version 0.38). Subsequently, the reads were aligned to the hg38 reference genome using bowtie 2 (v2.4.5).

To calculate the histone read coverage within each CGI (CpG island), we utilized the BEDtools coverage function. Initially, CGIs with less than 4 read counts in the ChIP-seq data were excluded to avoid including randomly mapped regions. The read counts were then adjusted to counts per 10 million based on the total number of mapped reads per sample. Additionally, the counts were divided by the input read count to normalize the read counts. To prevent normalized counts from becoming infinite in regions where the input sample had zero reads, an offset of 0.5 was added to all windows before scaling and input normalization. Regions where the coverage was zero in all samples were removed from the analysis.

In order to statistically analyze differences in histone modification levels, we compared the normalized read depths across CGIs using a Spearman's correlation tests. This test allowed us to assess the significance of differences in histone modification levels between samples.

## Chromatome analysis

We followed the protocol we have recently published[43]. All experiments were conducted in triplicates. After harvesting, $7 \times 10^6$ HCT116 cells for each condition were washed with PBS and split into $5 \times 10^6$ cells for downstream chromatin extraction and $2 \times 10^6$ for downstream whole cell lysis.

For the whole-cell proteome, the snap-frozen cell pellet was dissolved in 200 μL of lysis buffer (containing 6 M guanidinium chloride, 100 mM Tris-HCl with a pH of 8.5, and 2 mM DTT) and subjected to heating for 10 min at 99 °C with a constant agitation rate of 1400 rpm.

The sonication of samples was then performed at 4 °C using a Bioruptor® Plus for 15 cycles (30 s on, 30 s off). If the sample viscosity was adequately reduced, protein concentrations were determined; if not, sonication was repeated. Protein concentrations were assessed using the Pierce™ BCA Protein Assay Kit. After incubating for at least 20 min with 40 mM chloroacetamide, 30 µg of each proteome sample was diluted in a 50 µL lysis buffer supplemented with chloroacetamide and DTT. These samples were further diluted in 450 µL of digestion buffer (containing 10% acetonitrile, 25 mM Tris·HCl at pH 8.5, 0.6 µg Trypsin/sample, and 0.6 µg/sample LysC. Proteins were then digested for 16 h at 37 °C with constant shaking at 1100 rpm. To stop protease activity, 1% (v/v) trifluoroacetic acid (TFA) was added the following day and samples were loaded onto homemade StageTips composed of three layers of SDB-RPS matrix (Empore), previously equilibrated with 0.1% (v/v) TFA. After loading, two washes with 0.1% (v/v) TFA were performed, and peptides were eluted with 80% acetonitrile and 2% ammonium hydroxide. After the eluates were evaporated in a Speed-Vac centrifuge, the samples were resuspended in 20 µL 0.1% TFA and 2% acetonitrile. The peptides were completely solubilized by constant shaking for 10 min at 2000 rpm, and peptide concentrations were determined on a Nanodrop™ 2000 spectrophotometer (Thermo Fisher Scientific) at 280 nm.

For chromatome analysis, after two PBS washes, cells were resuspended in 1 mL of ice-cold lysis buffer (3 mM $MgCl_2$, 10 mM NaCl, 10 mM HEPES pH 7.4, 0.1% NP-40, and freshly added 1× cOmplete protease inhibitor (Roche)). The pellet was homogenized by pipetting up and down and then incubated on ice for 20 min. Crude nuclei were pelleterd at $2300 \times g$ for 5 min at 4 °C, and the supernatant was discarded. The pellet was resuspended and incubated in 1 mL of PBS with 1% methanol-free formaldehyde for 10 min on a rotating wheel at mild agitation at RT. The reaction was quenched by incubating the suspension with 125 mM Glycine for an additional 5 min on a rotating wheel. The nuclei suspension was then centrifuged at $2300 \times g$ for 5 min at 4 °C and washed twice with ice-cold PBS. After a second PBS wash, the supernatant was discarded, and the pellets were snap-frozen in liquid nitrogen for 15 s and then stored at −80 °C. Nuclei were lysed by adding 300 µL of SDS buffer (50 mM HEPES pH 7.4, 10 mM EDTA pH 8.0, 4% UltraPure™ SDS (Invitrogen) and freshly added 1× cOmplete™ EDTA-free Protease Inhibitor Cocktail) using gentle pipetting. This mixture was left to incubate at RT for 10 min before adding 900 µL of freshly prepared Urea buffer (10 mM HEPES pH 7.4, 1 mM EDTA pH 8.0, 8 M urea (Sigma)). The solution was then carefully inverted seven times before being centrifuged at RT for 30 min at $20,000 \times g$. The supernatant was removed, taking care not to disturb the pellet. Two additional wash steps were performed (one wash with SDS and Urea, and one wash with only SDS). The final pellet was then dissolved in 300 µL of Sonication buffer (10 mM HEPES pH 7.4, 2 mM $MgCl_2$ and freshly added 1× cOmplete™ EDTA-free Protease Inhibitor Cocktail). The chromatin samples were sonicated using a Bioruptor® Plus at 4 °C for 15 cycles (30 s on, 30 s off). The protein concentration was determined using the Pierce™ BCA Protein Assay Kit (Thermo Fisher Scientific) in accordance with the manufacturer's instructions. Subsequently, Protein Aggregation Capture (PAC) was performed. In this step, 1500 µg Sera-Mag™ beads (Sigma) per sample were washed three times with 70% acetonitrile for every 75 µg of chromatin solution. After the final wash, 300 µL of the chromatin solution corresponding to 75 µg was added to the beads, followed by 700 µL of 100% acetonitrile. The chromatome-bead mixtures were then vortexed and left to rest on a bench for 10 min. The samples were vortexed again and placed into a magnetic rack. The samples were then washed with 700 µL of 100% acetonitrile, followed by 1 mL of 95% acetonitrile, and finally with 1 mL of 70% ethanol. The remaining ethanol was allowed to evaporate, and the beads were resuspended in 400 µL of 50 mM HEPES pH 8.5, supplemented with freshly prepared 5 mM TCEP and 5.5 mM CAA. The samples were then left to incubate at RT for half an hour. Protease

digestion was initiated by adding LysC (protease to protein ratio of 1:200) and Trypsin (1:100) and allowing the mixture to incubate overnight at 37 °C under constant agitation at 1100 rpm. From this point forward, the samples were handled in the same way as the total proteome samples.

For all samples, peptide separation before MS was accomplished using liquid chromatography on an Easy-nLC 1200 (Thermo Fisher Scientific) with in-house packed 50 cm columns of ReproSilPur C18-AQ 1.9-µm resin. A binary buffer system was used (buffer A: 0.1% formic acid and buffer B: 0.1% formic acid in 80% acetonitrile), with a gradual increase in buffer B concentration (from 5% initially to 95% at the end) to elute the peptides over a 120-min period at a steady flow rate of 300 nL/min. The peptides were then introduced into an Orbitrap Exploris™ 480 mass spectrometer (Thermo Fisher Scientific) via a nanoelectrospray source. Each set of triplicates were followed by a washing step while the column temperature was constantly at 55 °C. Data-Independent Acquisition (DIA) runs used an orbitrap resolution of 120,000 for full scans in a scan range of 350–1400 $m/z$, with a maximum injection time of 45 ms. For MS2 acquisitions, the mass range was set to 361–1033 with isolation windows of 22.4 $m/z$. A default window overlap of 1 $m/z$ was used. The orbitrap resolution for MS2 scans was set at 30,000, the normalized AGC target at 1000%, and the maximum injection time at 54 ms.

Finally, raw MS data acquired in DIA mode was analyzed using DIA-NN version 1.8. Cross-run normalization was conducted in an RT-dependent manner. Missed cleavages were set to 1, N-terminal methionine excision was activated, and cysteine carbamidomethylation was set as a fixed modification. Proteins were grouped using the additional command "−relaxed-prot-inf." Match-between runs was enabled, and the precursor FDR was set to 1%. Mass accuracy was fixed to 2e-05 (MS2) and 5e-06 (MS1).

### Transfection and co-immunoprecipitation with GFP trap beads

In a 10 cm dish with HCT116 cells at 60% confluency, 12 micrograms of GFP-tagged plasmid (GFP, hUHRF1, UBL, TTD, PHD, SRA, RING, ΔUBL, ΔTTD, ΔPHD, ΔSRA, ΔRING, hUHRF1-TTD-mut) and 12 micrograms of dsRed-tagged plasmid (DNMT3A, DNMT3B) were transfected using 60 µL Lipofectamine 2000 (Thermo Fisher Scientific). After a 3-h incubation with Lipofectamine, the medium was replaced with McCoy's 5A medium and incubated for 1 day. The transfected cells were then collected by trypsinization, washed twice with PBS, and subjected to co-immunoprecipitation.

Co-immunoprecipitation was carried out following the manufacturer's protocol for GFP-Trap Agarose (chromotek). The collected cells were suspended in 200 µL lysis buffer (10 mM Tris/Cl pH 7.5, 150 mM NaCl, 0.5 mM EDTA, 0.5% NP-40, 2.5 mM $MgCl_2$, 1 mM PMSF, Protease inhibitor cocktail (Roche)) and incubated on ice for 30 min. The lysed samples were centrifuged at $16,000 \times g$ for 10 min at 4 °C. A portion of the supernatant was collected as input, and the remaining supernatant was combined with dilution buffer (10 mM Tris/Cl pH 7.5, 150 mM NaCl, 0.5 mM EDTA, 1 mM PMSF, Protease inhibitor cocktail (Roche)) to a final volume of 500 µL.

Subsequently, 30 µL of GFP-Trap Agarose, pre-equilibrated with dilution buffer, was added to each lysate sample. The samples were incubated overnight at 4 °C with gentle rotation. The GFP-Trap Agarose was then washed 5 times with lysis buffer and boiled for 10 min with SDS-PAGE sample buffer to elute the bound proteins for further analysis.

### RNA extraction and quantitative reverse transcription PCR (RT-qPCR)

Total RNA was extracted from cells with RNeasy Plus Mini kit (Qiagen) according to the manufacturer's instructions and quantified using Qubit RNA BR Assay kit on Qubit 2.0 Fluorometer (Thermo Fisher Scientific). For RT-qPCR, total RNA was reverse transcribed using

SuperScript IV Reverse Transcriptase (Thermo Fisher Scientific) and random primers (Promega). RT-qPCR was performed using Power SYBR Green (Applied Biosystems) following to manufacture protocol with TET2 and internal control (TBP1 and PGK1) primers. RT-qPCR primer sequences are available in Supplementary Table 1.

## Reporting summary

Further information on research design is available in the Nature Portfolio Reporting Summary linked to this article.

## Data availability

The WGBS and RNA-seq data has been submitted to GEO under references GSE236026 (WGBS, https://www.ncbi.nlm.nih.gov/geo/query/acc.cgi?acc=GSE236026) and GSE249536 (RNA-seq, https://www.ncbi.nlm.nih.gov/geo/query/acc.cgi?acc=GSE249536). The mass spectrometry proteomics data have been deposited to the ProteomeXchange Consortium via the PRIDE[70] partner repository with the dataset identifier PXD043254. H3K36me3 and H3K79me2 ChIP-seq data used in this study were obtained from ENCODE project (https://www.encodeproject.org/). The accession numbers are ENCSR091QXP (H3K36me3) and ENCSR494CCN (H3K79me2). Source data are provided with this paper.

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

## Acknowledgements

This work was supported by Agence Nationale de la Recherche (ANR-15-CE12-0012-01 to P.-A.D.; ANR-19-CE12-0030 to P.-A.D., H.L., and T.B.), Fondation ARC (ARC labellisation program 2019 to D.F.; PGA1 RF20180206807 to P.-A.D.; PDF20181208337 to K.Y.), Institut National du Cancer (INCa PLBio 2015-1-PLBio-01-DR A-1 to P.-A.D.), LabEx "Who Am I?" #ANR-11-LABX-0071 and the Université de Paris IdEx #ANR-18-IDEX-0001 (to P.-A.D and K.Y.), JSPS Overseas Research Fellowships (Ref. No. 202260432 to K.Y.), the Platform Project for Supporting Drug Discovery and Life Science Research (Basis for Supporting Innovative Drug Discovery and Life Science Research (BINDS)) from AMED under Grant Number JP20am0101103 (support number 2652, to F.M. and T.I.), the Deutsche Forschungsgemeinschaft (JE252/48 to A.J.), Foundation DFG (DFG Project ID 431163844 to D.H., B.O.A., H.L., and T.B.), and the DIMONEHEALTH 2019 equipment grant (project EpiK led by P.B.A. for the LC-MS to P.B.A. and F.B.). The authors acknowledge the Cell and Tissue Imaging (PICT-IBiSA), Institut Curie, member of the French National Research Infrastructure France-BioImaging (ANR10-INBS-04), Stéphanie Morchoisne, and the ImagoSeine core facility of Institut Jacques Monod, member of France-BioImaging (ANR-10-INBS-04), with the support of Plan Cancer, Region Ile-de-France and Fondation Bettencourt Schueller (R03/75-79). We also thank the Vectorology platform, Epigenetics platform, Microscopy platform and Bioinformatics/Biostatistics

Core Facility (BIBS) at the CNRS Epigenetics and Cell Fate Unit (Université Paris Cité), for providing access and technical advice. We are very grateful to Saadi Khochbin, Allison Bardin, Atsushi Kaneda, Raphael Margueron, and Adele Murrell for useful advice. We thank the following colleagues for the gift of useful reagents: Alexis Gautreau, Olivier Bernard, Michaela Fontenay.

## Author contributions

K.Y., X.C., B.R., and P.-A.D. designed and performed experiments; K.Y., X.C., B.R., A.S., and M.L. established cell lines and rescue constructs for this study. K.Y., X.C., B.R. (with the help of N.G., B.O.A., L.F., and M.L.) performed molecular biology experiments. K.Y., B.R., L.F., F.B. performed DNA methylation analysis (LUMA and LC-MS). C.S.-L. and D.H. performed microscopy analysis. F.M. prepared WGBS libraries, then sequenced them with NGS. P.B. and A.J. conducted the analyses shown in Figs. 4C and 6A. K.Y., P.B., F.M., O.K. performed bioinformatic analyses. E.U. performed proteome analysis. K.Y., P.B.A., H.L., M.T.K., D.F., A.J., T.I., T.B. and P.-A.D. acquired funding. K.Y. and P.-A.D. supervised the project. K.Y. wrote the original draft of the manuscript. A.J., D.F., B.R. and P.-A.D. reviewed and edited the draft.

## Competing interests

The authors declare no competing interests.
