## [Peer Review File · Nature Communications]

Non-canonical functions of UHRF1 maintain DNA methylation homeostasis in cancer cellsEditorial Note: Parts of this Peer Review File have been redacted as indicated to remove third-party material where no permission to publish could be obtained.

REVIEWER COMMENTS

Reviewer #1 (Remarks to the Author):

In this extensive manuscript, the authors explore the non-canonical roles of UHRF1 in colon cancer cells. They present 3 key findings, namely UHRF1 has roles beyond recruitment of DNMT1 to the replication fork to maintain DNA methylation during replication; UHRF1 interacts directly with DNMT3A and B; and UHRF1 antagonizes active DNA demethylation by TET2. The results are based on an elegant system to deplete UHRF1, DNMT1 or both in cells and study the DNA methylation kinetics, mostly at global levels, combined with a number of bioinformatic analysis.

Overall, the manuscript is well written, the data is nicely presented and an ample amount of complimentary experiments and analysis have been conducted to corroborate the findings in different settings.

Nevertheless, there are some points that require further attention, in particular to make the finding relevant in a broader context:

The whole work is based on cancer cells and it is important to clearly state this and also refer to the prior work in other systems, in particular stem cells and embryo where regulation of UHRF1 is also prominent and DNAm dynamics are high.

Introduction:

- The authors state that DNA demethylation in early embryo is a process "that involves active demethylation by the TET enzymes". This is in principle correct, but it should be said that passive loss of DNAm is likely the key contributor (PGCs: Surani et al.; Embryo: Reik et al., including a MS specifically on UHRF1 and DNMT1 during ESC progression)
- It might be good to briefly mention DNMT3C (Barau et al 2016)
- UHRF1 "binds histone H3K9me3". It might be correct to say H3K9me2/3 for completeness
- Overall, it would be good to briefly discuss the existing work on DNAm dynamics in the existing other studies, e.g. in ESCs, embryo, germ cells, Aza treatment...

Results

- Fig1: The effects on DNAm levels and proliferation by "just" having the AID tag are a bit worrying. How can the authors exclude that in the subsequent results these pre-existing changes are not causal for the effects observed?
- Fig2: Considering Fig1, the authors should aim to consider if the effects of the cell line itself can be ignored. In particular since the effects (without Auxin) seems to be reproducible
- Fig2: The effects observed in ESCs are not 100% overlapping and this should be taken into account. The degron system is of course superior to loxP strategies, but might introduce other unknown effects.
- Fig2: How do the authors interpret the differences in degradation between UHRF1 or DNMT1 only and combined cell lines? It seems in the UHRF1/DNMT1 experiments, the proteins levels remain detectable for at least 8-24hrs
- Fig3: How well does the demethylation compare to AZA treatment or other experiments done before. Do the kinetics fit to a complete loss of DNAm maintenance?
- Fig 3C: are these single replicates or pooled samples?
- Fig 4&5: The interaction with DNMT3a/b is interesting, but how the authors explain the stronger correlation with DNMT3a then b sites, but in Fig5 a similar binding? Also, despite the various (good) attempts to exclude chromatin binding as a cause of the Co-IPs, interaction with DNA or overall high affinity for similar sites can not be excluded. Did the authors consider also checking for other chromatin binders to exclude that the binding is unspecific.
- Fig6: The data on TET2 is also interesting, but given the small differences shown in Fig1, it is not clear how strong the cell line only effect can be separated from the Tet2 phenotype. Maybe targeted recruitment of UHRF1 to specific genomic sites which are then protected from 5hmC generation would be a way to better show this.

General Feedback on Stats: It would be better to show SD instead of SEM for the small sample numbers.

Reviewer #2 (Remarks to the Author):

The work of Yamaguchi et al. reveals non-canonical roles for UHRF1 in the maintenance of DNA methylation. UHRF1 has initially been identified as a critical factor for the maintenance of DNA methylation during DNA replication. Later, it was shown that UHRF1 regulates DNA methylation by recruiting DNMT1 to DNA methylation sites. In this study, the authors generated degron alleles of UHRF1 and/or DNMT1 in human colorectal cancer cell lines. The first part of the work confirmed that UHRF1 and/or DNMT1 depletion leads to growth inhibition and DNA methylation loss. The authors also performed genetic rescue experiments by expressing DNMT1 and UHRF1 variants to identify the functional domains supporting cell growth and maintaining DNA methylation. Importantly, they found that UHRF1 depletion causes a more severe DNA methylation defect than DNMT1 depletion, particularly in gene bodies. Since it is well-known that the de novo DNA methyltransferases DNMT3A and DNMT3B are involved in gene-body methylation, the authors next tested the physical and genetic interaction between UHRF1 and de novo DNA methyltransferases. They found that UHRF1 interacts with DNMT3A and 3B via the TTD domain of UHRF1. Furthermore, knockouts of DNMT3A and DNMT3B in UHRF1-AID lines did not enhance the loss of DNA methylation, suggesting that UHRF1 acts upstream of DNMT3A and DNMT3B. They further demonstrated that TET2 activity contributes to DNA methylation loss in UHRF1-depleted cells. These findings collectively support that UHRF1 is essential in maintaining DNA methylation, regulating a variety of DNA methylation regulators.

The manuscript is very interesting, concise, clearly written, and have broad implication for our understanding of DNA methylation inheritance. I have a few issues that I would like the authors to consider before publication:

Major points

1) In Fig. 2C and S2C, the authors showed that DNMT1- or UHRF1-depletion leads to growth defect without cell death. Is this due to cell cycle arrest at a specific phase?

2) The authors showed that UHRF1 depletion leads to decreased DNA methylation at DNMT3A and DNMT3B target sites, in addition to DNMT1 target sites. Is this due to defective recruitment of DNMT3A/B to DNA methylation sites?

3) Although DNMT3A and DNMT3B interact with the hydrophobic pocket of the TTD domain of UHRF1 (Figure 5B and C), the TTD mutant efficiently supports DNA methylation (Fig. S3B). How do the authors explain this result?

4) Fig. 6 shows TET2 activity promotes DNA methylation loss in the absence of UHRF1, suggesting that UHRF1 negatively regulates TET2. However, the underlying molecular mechanism has not been demonstrated. For example, could the authors test whether 5mc oxidation occurs at the TET2 target site in UHRF1-depleted cells?

Minor issues:

1) Introduction: In addition to ubiquitylation-dependent mechanisms, direct interactions between UHRF1 and DNMT1 have been reported in many studies and should be cited.

2) Major mono-ubiquitylation sites of histone H3 are H3K18/K23, not K14/K18.

Reviewer #3 (Remarks to the Author):

The maintenance cytosine methyltransferase, DNMT1, is required to perpetuate DNA methylation patterns that are programmed in development and in somatic, pluripotent and cancer cells. However, for this function it requires an array of accessory proteins including de novo methyltransferases, the UHRF proteins and chromatin remodelling complexes. Thus, mouse ES cells lacking de novo methyltransferases become severely hypomethylated during passaging as Dnmt1 is unable to maintain patterns by itself.

As noted by the authors, many cancer cells have altered DNA methylation profiles compared to their cognate somatic counterparts. HCT1116 cells, used here, are hypomethylated and have altered landscapes including PMD's (partially methylated domains), indicating altered dependency by cancer cells on the DNA methylation machineries. This can create complications when studying functional aspects of DNA methylation programming. In the past, DNMT1 mutants in HCT116 have turned out to be hypomorphs, exhibiting partial hypomethylation that can be enhanced more when DNMT3b is inactivated (a 95% reduction in m5C content in representative DKO clones). Previously data from the Leonhardt lab reported that RNA interference-mediated knockdown of DNMT1 hypomorphic HCT116 cells results in global genomic hypomethylation and cell death.

The results of the genetic inactivation experiments represent a potential divergence when using the Auxin-Inducible Degron (AID) system to perform DNMT1 and UHRF1 loss-of-function experiments. The two systems are not equivalent as loss of DNMT1-AID function results in partial hypomethylation (perhaps 25%) and loss of DNMT1-AID function on a DNMT3A/3B KO background does not result in complete hypomethylation as commented non-quantitatively by the authors - 'led to a greater loss of DNA methylation upon auxin treatment (Fig. 5E)'.

What's the explanation? The obvious one is that AID mediated loss of function is incomplete, which is hard to assess by a limit of detection assay (western blot) which is non-quantitative. An obvious way to resolve this is to compare the functional consequence of a mega CRISPR mediated KO of the complete DNMT1 locus in HCT116 cells with AID mediated DNMT1 inactivation, likely not to be equivalent. However, if the comparative experiments yield the same results, the presence of significant levels of DNA methylation, then why? Is it unexplained new methyltransferase activities in these cells or perhaps genetic re-arrangements of the DNMT loci resulting in additional functional gene copies? This needs to be clarified before functional interpretations of the AID mediated reduction in DNMT1 or UHRF1 activities.

Other comments:

The cell cycle patterns and co-localisation of the tagged DNMT1 and UHRF1 proteins in HCT116 cells needs more extensive investigation and analysis as performed in the past to a high degree of excellence by the Leonhardt lab. What are the sub-nuclear foci tagged by DNMT1 and UHRF1 for example, human chromosomes with significant heterochromatin?

The cell number analysis needs to be complemented by potential DEG expression analysis in the depleted DNMT1, UHRF1 and (DNMT1/UHRF1) cells to allow for the potential contribution of DEG to altered cellular properties, including cell cycle changes.

The functional interaction IP studies of over-expressed proteins should be complemented by IP analysis of endogenous WT proteins.

The chromatin protocols and results need a much fuller explanation.

The TET2 experiments need a genetic inactivation counterpart with LC-MS/MS analysis.

Paris december 7th 2023

Yamaguchi et al NCOMMS-23-27402 Rebuttal letter

We are grateful to the three reviewers for their insightful comments; they have been duly taken account and have helped us improve the manuscript. The comments are copied verbatim below in black, with our responses in blue.

Reviewer #1

In this extensive manuscript, the authors explore the non-canonical roles of UHRF1 in colon cancer cells. They present 3 key findings, namely UHRF1 has roles beyond recruitment of DNMT1 to the replication fork to maintain DNA methylation during replication; UHRF1 interacts directly with DNMT3A and B; and UHRF1 antagonizes active DNA demethylation by TET2. The results are based on an elegant system to deplete UHRF1, DNMT1 or both in cells and study the DNA methylation kinetics, mostly at global levels, combined with a number of bioinformatic analysis.

> Thank you for this accurate summary of the manuscript.

Overall, the manuscript is well written, the data is nicely presented and an ample amount of complimentary experiments and analysis have been conducted to corroborate the findings in different settings.

> We are grateful for your positive assessment of our work.

Nevertheless, there are some points that require further attention, in particular to make the finding relevant in a broader context:

The whole work is based on cancer cells and it is important to clearly state this and also refer to the prior work in other systems, in particular stem cells and embryo where regulation of UHRF1 is also prominent and DNAm dynamics are high.

> Yes we agree. To accommodate your request, we have modified several paragraphs of the discussion, and compared our results in cancer cells to the situation in other biological systems, including stem cells and the embryo.

Introduction:

- The authors state that DNA demethylation in early embryo is a process “that involves active demethylation by the TET enzymes”. This is in principle correct, but it should be said that passive loss of DNAm is likely the key contributor (PGCs: Surani et al.; Embryo: Reik et al., including a MS specifically on UHRF1 and DNMT1 during ESC progression)

> Thank you for pointing this out. This clarification has been made. The number of references we are permitted is limited at 70, so instead of several primary papers, we now direct the reader towards an authoritative review by Wolf Reik that deals specifically with this question (Lee et al, Cell Stem Cell 2014).

- It might be good to briefly mention DNMT3C (Barau et al 2016)

> This has been added, thank you.

- UHRF1 “binds histone H3K9me3”. It might be correct to say H3K9me2/3 for completeness

> Absolutely, this has been amended.

- Overall, it would be good to briefly discuss the exciting work on DNAm dynamics in the existing other studies, e.g. in ESCs, embryo, germ cells, Aza treatment...

> Thank you for this valuable comment. As stated above, in response to an earlier comment of yours, this has now been done in the discussion. In addition, we have now performed an experiment with 5-aza, as you suggested (new Figs 3D-E, p.4 of this document).

Results

- Fig1: The effects on DNAm levels and proliferation by “just” having the AID tag are a bit worrying. How can the authors exclude that in the subsequent results these pre-existing changes are not causal for the effects observed?

> Thank you for bringing up this important point, which is indeed a very important factor to control. We apologize if this was not clear enough, but figures 1 and S1 are precisely dedicated to determining what the effect of just having the AID tag is. When we look at DNA methylation, by three different methods, we find that there is no significant difference between HCT116 WT, UHRF1-AID, and DNMT1-AID cells (Fig. 1E). The combined tagged line UHRF1-AID/DNMT1-AID does have a minor defect, but that does not in any way affect our conclusions on the single-tag lines. When we look at proliferation (Fig. S1CD) the differences between HCT116 WT, UHRF1-AID, and DNMT1-AID are, again, extremely minor. Finally, as a measure to assuage these concerns, we have now carried out RNA-seq, as described in the answer to your next question.

- Fig2: Considering Fig1, the authors should aim to consider if the effects of the cell line itself can be ignored. In particular since the effects (without Auxin) seems to be reproducible

> We are grateful for this suggestion, which parallels the point you have raised above. It is indeed a possibility that the cell line itself has effects, as suggested by the observation that the effects (without Auxin) are reproducible. We have considered this in depth, and we could not come up with additional ways to mitigate this risk, besides what is already shown in the paper, namely:

- carrying out all experiments with or without auxin
- using three independent biological clones of each line
- examining 2 different lines
- performing rescue experiments

In the revised version of the manuscript, we have taken one additional step to characterize the effects of the tag: we have carried out RNA-seq on all lines in the absence of auxin (see new figure S1G, copied below). Prior to auxin treatment, there is virtually no detectable transcriptional difference between UHRF1-AID and WT, or between DNMT1-AID and WT. In contrast, the compound UHRF1-AID/DNMT1-AID line does differ from WT, with 124 genes down, and 132 genes up, again in the absence of auxin. This deregulation likely stems from the slightly altered DNA methylation in the compound line. Please note, however, that the number of genes affected is small in comparison to the effect seen after auxin addition (320 genes down, 1787 genes up, new figure S2E).

New Figure S1G: RNA-seq analysis on the indicated lines in the absence of auxin

- Fig2: The effects observed in ESCs are not 100% overlapping and this should be taken into account. The degron system is of course superior to loxP strategies, but might introduce other unknown effects.

>Yes you are right. We now consider this possibility in the discussion section.

- Fig2: How do the authors interpret the differences in degradation between UHRF1 or DNMT1 only and combined cell lines? It seems in the UHRF1/DNMT1 experiments, the proteins levels remain detectable for at least 8-24hrs

> Thank you for this perceptive comment. You are absolutely right: in the cells where only UHRF1 or only DNMT1 is tagged, the degradation is swift and complete. In the cells where both proteins are tagged simultaneously, the degradation is slower and possibly incomplete (Figure 2A).

We have noticed this as well, and before submitting the paper, we have ruled out a first possible explanation. It could just have been that, in the combined cell lines, a fraction of the cells did not express OsTIR1 any longer, preventing degradation in these cells. However, the silencing or loss of OsTIR1 should be accompanied by the loss of the adjacent selection marker, PuroR. Therefore, we resubmitted the cells to a round of Puromycin selection. This had no effect whatsoever on the degradation pattern, and we can therefore rule out the loss of OsTIR1 expression as an explanation.

Another possibility would be that the system is overloaded when the 2 proteins have to be degraded. In other words, there is enough machinery in the cell to degrade UHRF1 alone, or to degrade DNMT1 alone, but not enough machinery to degrade both. This possibility is supported by published data: in Macdonald et al, Elife 2022, PMID: 35736539, it is convincingly shown that a high ratio of OsTIR1 to target proteins is important to ensure prompt and total degradation (Figures 3 and 4 of that paper). We now mention this article in our manuscript.

- Fig3: How well does the demethylation compare to AZA treatment or other experiments done before. Do the kinetics fit to a complete loss of DNAm maintenance?

> This is an excellent question, thank you for giving us the chance to expand on this. As you suggested, we have compared the effect of AZA to that of UHRF1 or DNMT1 depletion (see new figures 3D-E, below). The takeaway is that UHRF1 depletion mimics AZA treatment, whereas DNMT1 depletion has a milder effect. We explain this by the fact that AZA degrades not only DNMT1, but also DNMT3A and DNMT3B, as has been reported (PMID: 15735669, PMID: 20348135, PMID: 33859525), and as we verified in our cells.

New Figures 3D-E: Effect of a 5-aza treatment on DNA methylation and protein abundance

- Fig 3C: are these single replicates or pooled samples?

> The results shown in Fig 3C are the aggregates of three independent clones. The results for each individual clones are shown below, for your perusal.

New Figure S3A: WGBS data for individual clones

Uaid: UHRF1-AID; Daid: DNMT1-AID; UD: UHRF1-AID/DNMT1-AID

Y-axis: Percentage CpG methylation; X-axis: days of auxin treatment

- Fig 4&5: The interaction with DNMT3a/b is interesting, but how the authors explain the stronger correlation with DNMT3a than b sites, but in Fig5 a similar binding?

> Thank you for this discerning question. At first sight, it may indeed seem surprising that we see similar binding to DNMT3A and DNMT3B (Fig. 5), but a stronger correlation with DNMT3A than DNMT3B sites. However, it may be difficult to directly compare these two experiments, as two additional factors would have to be considered in the equation: first, the relative amount of UHRF1, DNMT3A, and DNMT3B and, second, the relative activity of DNMT3A and DNMT3B, which itself depends on the isoforms that are expressed. Therefore, again, we would be careful about drawing strong parallels between the two types of experiments.

Also, despite the various (good) attempts to exclude chromatin binding as a cause of the Co-IPs, interaction with DNA or overall high affinity for similar sites can not be excluded. Did the authors consider also checking for other chromatin binders to exclude that the binding is unspecific.

> Thank you for this interesting suggestion. As you say, it is useful to control further that, when we IP UHRF1, we do not just pull down chromatin fragments that are enriched in various chromatin binders, including DNMT3A and DNMT3B. To address this possibility, we have carried out new co-IPs and tested whether the immunoprecipitates contain other chromatin binders. Specifically, we have looked for HP1 α , β , and γ , which together should cover a large part of the genome. As you can see in the new Reviewer Figure below, the UHRF1 immunoprecipitate contains DNMT3A and DNMT3B, but it does not contain HP1 α , β , or γ . This argues against an indiscriminate pull-down of chromatin fragments, and in favor of a specific co-IP of UHRF1, DNMT3A and DNMT3B.

Reviewer Figure 1: HP1 proteins do not co-IP with UHRF1
because of space constraints, this figure does not appear in the MS

- Fig6: The data on TET2 is also interesting, but given the small differences shown in Fig1, it is not clear how strong the cell line only effect can be separated from the Tet2 phenotype. Maybe targeted recruitment of UHRF1 to specific genomic sites which are then protected from 5hmC generation would be a way to better show this.

> Please accept my apologies, but I am not quite sure I understand this suggestion. Provided there is no typo, you are referring to Fig1, in which we do examine different cell types (HCT116, DLD1), but all is done in the absence of auxin. In other words, in this figure, we only determine whether the tags have an effect in themselves, and whether the cell types differ in basal conditions. But from Fig1 we cannot say anything as to the dynamics of demethylation, and therefore we cannot draw any conclusion as to how much (or how little) TET2 contributes to DNA demethylation.

General Feedback on Stats: It would be better to show SD instead of SEM for the small sample numbers.

> Absolutely, thanks for pointing this out. This correction has been done throughout the manuscript.

Reviewer #2

The work of Yamaguchi et al. reveals non-canonical roles for UHRF1 in the maintenance of DNA methylation. UHRF1 has initially been identified as a critical factor for the maintenance of DNA methylation during DNA replication. Later, it was shown that UHRF1 regulates DNA methylation by recruiting DNMT1 to DNA methylation sites. In this study, the authors generated degron alleles of UHRF1 and/or DNMT1 in human colorectal cancer cell lines. The first part of the work confirmed that UHRF1 and/or DNMT1 depletion leads to growth inhibition and DNA methylation loss. The authors also performed genetic rescue experiments by expressing DNMT1 and UHRF1 variants to identify the functional domains supporting cell growth and maintaining DNA methylation. Importantly, they found that UHRF1 depletion causes a more severe DNA methylation defect than DNMT1 depletion, particularly in gene bodies. Since it is well-known that the de novo DNA methyltransferases DNMT3A and DNMT3B are involved in gene-body methylation, the authors next tested the physical and genetic interaction between UHRF1 and de novo DNA methyltransferases. They found that UHRF1 interacts with DNMT3A and 3B via the TTD domain of UHRF1. Furthermore, knockouts of DNMT3A and DNMT3B in UHRF1-AID lines did not enhance the loss of DNA methylation, suggesting that UHRF1 acts upstream of DNMT3A and DNMT3B. They further demonstrated that TET2 activity contributes to DNA methylation loss in UHRF1-depleted cells. These findings collectively support that UHRF1 is essential in maintaining DNA methylation, regulating a variety of DNA methylation regulators.

> Thank you for this accurate summary of the manuscript.

The manuscript is very interesting, concise, clearly written, and have broad implication for our understanding of DNA methylation inheritance.

> We are very thankful for your support of our work.

I have a few issues that I would like the authors to consider before publication:

Major points

1) In Fig. 2C and S2C, the authors showed that DNMT1- or UHRF1-depletion leads to growth defect without cell death. Is this due to cell cycle arrest at a specific phase?

> Thank you for this recommendation. As you suggested, we have investigated this question, using FACS. As shown in the new Fig S2D, copied below, the cells do accumulate in G1 phase after 4 days of auxin exposure. Also, the sub-G1 population is barely detectable, in line with the absence of cell death.

New Figure S2D: FACS analysis of the indicated cell lines 4 days after auxin addition

2) The authors showed that UHRF1 depletion leads to decreased DNA methylation at DNMT3A and DNMT3B target sites, in addition to DNMT1 target sites. Is this due to defective recruitment of DNMT3A/B to DNA methylation sites?

> Thank you for giving us the chance to clarify this. The chromatome and western blot results of Figures 5 and S5 show that there is less DNMT3B on chromatin and, more generally, in the cell. We believe this contributes to some of the effects we are seeing.

3) Although DNMT3A and DNMT3B interact with the hydrophobic pocket of the TTD domain of UHRF1 (Figure 5B and C), the TTD mutant efficiently supports DNA methylation (Fig. S3B). How do the authors explain this result?

> This is a very perceptive point, as there is indeed an apparent conundrum. We believe the explanation is the following: a key function of UHRF1 is to promote DNMT1 activity, and that is (mostly) TTD-independent. This function is quantitatively preeminent, thereby masking the DNMT3A/B-dependent role. In order to uncover the latter role, we have re-introduced UHRF1 (WT or TTD-mutated) in cells that have neither UHRF1 nor DNMT1. In that experimental situation, depicted in the figure below, we observe that a UHRF1 construct mutated in the TTD is not as efficient at rescuing the cell line as a WT UHRF1. Therefore, we unmask a DNA methylation-promoting function of the TTD when DNMT1 is absent, which is consistent with our other results.

New Figure S5E: A TTD mutant form of UHRF1 only partially rescues UHRF1-AID/DNMT1-AID cells

4) Fig. 6 shows TET2 activity promotes DNA methylation loss in the absence of UHRF1, suggesting that UHRF1 negatively regulates TET2. However, the underlying molecular mechanism has not been demonstrated. For example, could the authors test whether 5mC oxidation occurs at the TET2 target site in UHRF1-depleted cells?

> Thank you for this very pertinent remark. In spite of our efforts, we have not been able to address this question within the timeline of this revision. We hope you will appreciate that almost all other questions by yourself and the other reviewers have been answered experimentally, and that further mechanistic understanding would not alter our message and could be kept for a follow-up study.

Minor issues:

1) Introduction: In addition to ubiquitylation-dependent mechanisms, direct interactions between UHRF1 and DNMT1 have been reported in many studies and should be cited.

> Absolutely. We apologize for the oversight, and we now cite in the introduction 4 of the key papers that have reported a direct interaction between UHRF1 and DNMT1: Achour Oncogene 2008 (Bronner lab, PMID: 17934516); Felle NAR 2011 (Längst lab, PMID: 21745816); Bashtrykov JBC 2014 (Jeltsch lab, PMID: 24368767); Berkyurek JBC 2014 (Tajima lab, PMID: 24253042).

2) Major mono-ubiquitylation sites of histone H3 are H3K18/K23, not K14/K18.

> We apologize for this mistake, which has been corrected.

Reviewer #3

The maintenance cytosine methyltransferase, DNMT1, is required to perpetuate DNA methylation patterns that are programmed in development and in somatic, pluripotent and cancer cells. However, for this function it requires an array of accessory proteins including de novo methyltransferases, the UHRF proteins and chromatin remodelling complexes. Thus, mouse ES cells lacking de novo methyltransferases become severely hypomethylated during passaging as Dnmt1 is unable to maintain patterns by itself.

As noted by the authors, many cancer cells have altered DNA methylation profiles compared to their cognate somatic counterparts. HCT116 cells, used here, are hypomethylated and have altered landscapes including PMD's (partially methylated domains), indicating altered dependency by cancer cells on the DNA methylation machineries. This can create complications when studying functional aspects of DNA methylation programming. In the past, DNMT1 mutants in HCT116 have turned out to be hypomorphs, exhibiting partial hypomethylation that can be enhanced more when DNMT3b is inactivated (a 95% reduction in m5C content in representative DKO clones). Previously data from the Leonhardt lab reported that RNA interference-mediated knockdown of DNMT1 hypomorphic HCT116 cells results in global genomic hypomethylation and cell death.

> Thank you for this concise and accurate summary of the salient published data. We wholeheartedly agree about the limitations of the HCT116 system. And thank you for bringing up this previous paper of the Leonhardt lab, Spada et al, JCB 2007, PMID: 17312023. I have copied part of Figure 3 of this paper below, as it is relevant to our discussion.

[REDACTED]

Figure 3 from Spada et al, JCB 2007.

These data show that, even in severely hypomethylated cells (DNMT1 hypomorph or DKO + 12 days RNAi), proliferation still occurs (panel B), and some cell death is observed, yet the lethality is partial (panel C).

The results of the genetic inactivation experiments represent a potential divergence when using the Auxin-Inducible Degron (AID) system to perform DNMT1 and UHRF1 loss-of-function experiments. The two systems are not equivalent as loss of DNMT1-AID function results in partial hypomethylation (perhaps 25%) and loss of DNMT1-AID function on a DNMT3A/3B KO background does not result in complete hypomethylation as commented non-quantitatively by the authors - 'led to a greater loss of DNA methylation upon auxin treatment (Fig. 5E)'.

> Thank you for this insightful discussion. As you rightfully point out, our results do not appear at first sight to match well with the results published by Ina Rhee and Bert Vogelstein in their landmark 2000 and 2002 Nature papers. For this reason we have compared our cells to theirs (DNMT1 hypomorph combined with DNMT3b inactivation, or "DKO"). We obtained the cells directly from the Vogelstein lab some years ago, and got them out of storage for comparison experiments.

First, we performed LUMA, a restriction-based assay (see below). This assay measures the percentage of restriction-resistant sites, i.e. those that are methylated or hemimethylated. In this assay, we got similar values for cells with a 12-day depletion of UHRF1 or UHRF1/DNMT1 and for the DKO cells (please see below). The DNMT1 degron cells did not quite reach the same level of depletion, as may have been expected.

Revised Figure 3A. LUMA experiment comparing the degron lines to DKO cells.

The LUMA assay only gives an indirect estimation of global DNA methylation, so we also performed shallow WGBS on the WT and DKO cells, and compared them to our results. The results were the following:

- our HCT116 (from the ATCC) and the Vogelstein HCT116 were reassuringly similar
- the DNMT1 depleted cells did not reach the same degree of methylation loss as the DKO
- but the UHRF1 depleted or UHRF1/DNMT1 depleted cells actually reached similar or greater degrees of methylation loss than the DKO.

Revised Figure 3C. WGBS experiment comparing the degron lines to DKO cells

Therefore, our long-term depleted cells and the Vogelstein DKO have similar levels of methylation loss. However, there is a major difference between these cells, which is relevant to the next discussion point: our depleted cells display extremely slow growth and an accumulation in G1, while the DKO cells proliferate as well as WT cells. The simplest interpretation is that the DKO cells have adapted to their low DNA methylation level, by genetic and/or epigenetic means.

What's the explanation? The obvious one is that AID mediated loss of function is incomplete, which is hard to assess by a limit of detection assay (western blot) which is non-quantitative. An obvious way to resolve this is to compare the functional consequence of a mega CRISPr mediated KO of the complete DNMT1 locus in HCT116 cells with AID mediated DNMT1 inactivation, likely not to be equivalent. However, if the comparative experiments yield the same results, the presence of significant levels of DNA methylation, then why? Is it unexplained new methyltransferase activities in these cells or perhaps genetic rearrangements of the DNMT loci resulting in additional functional gene copies? This needs to be clarified before functional interpretations of the AID mediated reduction in DNMT1 or UHRF1 activities.

> I respect your obvious knowledge of the field, but I humbly disagree that CRISPR mutations of DNMT1 would be feasible. Based on our data, in all likelihood, DNMT1 loss-of-function cells would be severely growth-impaired, and may even fail to form colonies. At any rate, there would be strong positive selection for genetic or epigenetic compensation.

Other comments:

The cell cycle patterns and co-localisation of the tagged DNMT1 and UHRF1 proteins in HCT116 cells needs more extensive investigation and analysis as performed in the past to a high degree of excellence by the Leonhardt lab. What are the sub-nuclear foci tagged by DNMT1 and UHRF1 for example, human chromosomes with significant heterochromatin?

> Thank you for encouraging us to develop this aspect of the work. We asked our collaborator Heinrich Leonhardt to supervise these experiments in his team, as he is indeed an expert. The data are shown below: UHRF1 and DNMT1 do colocalize. In addition, they colocalize with a marker of heterochromatin, H3K9me3.

New Figures 1D and S1F. Colocalization analyses of UHRF1, DNMT1, and H3K9me3.

The cell number analysis needs to be complemented by in the depleted DNMT1, UHRF1 and (DNMT1/UHRF1) cells to allow for the potential contribution of DEG to altered cellular properties, including cell cycle changes.

> Thank you for giving us the chance to expand on this important question. As you suggested, we have performed RNA-seq and done a DEG expression analysis in the depleted DNMT1, UHRF1 and (DNMT1/UHRF1) cells, 4 days after auxin was added (New Fig. S2E-G, copied below). The data are informative, and consistent with the rest of our study. Indeed we find that:

- more genes are deregulated upon UHRF1 depletion than upon DNMT1 depletion
- cell cycle genes, such as E2F targets, are among the downregulated genes in UHRF1-depleted and DNMT1-depleted cells.
- genes known to be repressed by DNA methylation, such as germline genes, are upregulated

New Figure S2E-G: RNA-seq and DEG analysis after 4 days of auxin treatment, comparing the indicated pairs of samples. Downregulated genes in blue, upregulated genes in red.

The functional interaction IP studies of over-expressed proteins should be complemented by IP analysis of endogenous WT proteins.

> We thank you for raising this important point. As you requested, we have carried out endogenous co-IPs of UHRF1, DNMT3A, and DNMT3B. The results are shown in the new Figure 5X, reproduced below for your convenience. We used the cells in which endogenous UHRF1 bears the mAID-mClover tag (or WT HCT116 cells as a negative control). We then IP'ed this endogenous UHRF1-mAID-mClover protein using anti-GFP beads, which also react against mClover. Lastly, we tested whether the immunoprecipitate contained DNMT3A or DNMT3B. As you can see below, it did, further supporting the interaction between UHRF1 and the de novo methyltransferases.

New Figure 55C-D: Co-immunoprecipitation of endogenous UHRF1, DNMT3A, and DNMT3B

The chromatome protocols and results need a much fuller explanation.

->The protocols and results have been expanded as you suggested, thank you.

The TET2 experiments need a genetic inactivation counterpart with LC-MS/MS analysis.

->In spite of our best efforts, we were unable to provide the genetic inactivation counterpart for the TET2 experiment. We would like to point out that the shTET2 we used has been validated in an independent publication (Uribe-Lewis Genome Biol 2015, PMID: 25853800). In addition, and to follow your recommendation, we have carried out the LC-MS/MS analysis for the TET2 experiment (please see new figure 6D, below).

New Figure 6D: LC-MS/MS analysis of the TET2 experiments

REVIEWERS' COMMENTS

Reviewer #1 (Remarks to the Author):

The authors have carefully revised the manuscript and addressed my concerns. I believe the manuscript has been improved and can now be published

Reviewer #2 (Remarks to the Author):

None

Reviewer #3 (Remarks to the Author):

The authors have done a very fine job in addressing all the comments by the referee's, and essentially promoting a genetic/dynamic interpretation for the consequences of aberrant DNA methylation patterns in cancer. The direct 'epigenetic changes' seem trivial in comparison to the non-canonical functions (signalling pathways) of the components of the DNA methylation machineries that they have identified.

The rebuttal was interesting. Clearly this paper will promote a lot of discussion amongst the aficionado's of DNA methylation, and in doing so nudge the field forward from out-dated assumptions that can hold back interpretation.

Well done to all.

REVIEWERS' COMMENTS

Reviewer #1 (Remarks to the Author):

The authors have carefully revised the manuscript and addressed my concerns. I believe the manuscript has been improved and can now be published

Reviewer #2 (Remarks to the Author):

None

Reviewer #3 (Remarks to the Author):

The authors have done a very fine job in addressing all the comments by the referee's, and essentially promoting a genetic/dynamic interpretation for the consequences of aberrant DNA methylation patterns in cancer. The direct 'epigenetic changes' seem trivial in comparison to the non-canonical functions (signalling pathways) of the components of the DNA methylation machineries that they have identified.

The rebuttal was interesting. Clearly this paper will promote a lot of discussion amongst the aficionado's of DNA methylation, and in doing so nudge the field forward from out-dated assumptions that can hold back interpretation.

Well done to all.

> We thank the three reviewers for accepting the papers without further changes